# The Pick-to-Learn Algorithm: Empowering Compression for Tight Generalization Bounds and Improved Post-training Performance

**Dario Paccagnan**
Dept. of Computing
Imperial College London
d.paccagnan@imperial.ac.uk

**Marco C. Campi**
Dip. di Ingegnerira dell'Informazione
Università di Brescia
marco.campi@unibs.it

**Simone Garatti**
Dip. di Elettronica Informazione e Bioingegneria
Politecnico di Milano
simone.garatti@polimi.it

## Abstract

Generalization bounds are valuable both for theory and applications. On the one hand, they shed light on the mechanisms that underpin the learning processes; on the other, they certify how well a learned model performs against unseen inputs. In this work we build upon a recent breakthrough in *compression theory* (Campi & Garatti, 2023) to develop a new framework yielding tight generalization bounds of wide practical applicability. The core idea is to embed any given learning algorithm into a suitably-constructed meta-algorithm (here called Pick-to-Learn, P2L) in order to instill desirable compression properties. When applied to the MNIST classification dataset and to a synthetic regression problem, P2L not only attains generalization bounds that compare favorably with the state of the art (test-set and PAC-Bayes bounds), but it also learns models with better post-training performance.

## 1 Introduction

Machine learning has achieved remarkable results in the last decades, with successful stories ranging from digit recognition (Dosovitskiy et al., 2020), to protein folding (Jumper et al., 2021), traffic prediction (Li et al., 2017), medical diagnosis (Kononenko, 2001), and beyond. Contrary to that, the understanding of the mechanisms underpinning generalization, and the availability of bounds for its evaluation, is still limited. This is unfortunate because generalization bounds are important both to develop trust in learning methods as well as to allow for a fair comparison among alternatives, which is relevant to hyper-parameter tuning.

Among the various tools that have been introduced to provide generalization bounds, we recall the VC-dimension (Vapnik & Chervonenkis, 1971), Radamacher complexity (Bartlett & Mendelson, 2002), mutual information (Xu & Raginsky, 2017), sharpness (Keskar et al., 2017), compression schemes (Littlestone & Warmuth, 1986; Floyd & Warmuth, 1995; Graepel et al., 2005), and PAC-Bayes approaches (Dziugaite & Roy, 2017; Perez-Ortiz et al., 2021). Each of these frameworks is applicable to specific contexts. For example, the approach based on the VC-theory requires that a model is selected from a class with finite VC-dimension, while compression schemes are applicable provided the existence of an informative subset of data points from which the model can be reconstructed. It has also to be said that the level of precision of the available bounds is highly problem-dependent and they often take a significant margin from the actual performance.

37th Conference on Neural Information Processing Systems (NeurIPS 2023).

While bounds based on the VC-dimension, and some other aforementioned approaches, are often conservative (resulting in untight, or even vacuous, evaluations), the sharpest approach to evaluate generalization across various learning domains is still that based on test-set bounds (Langford, 2005). However, the test-set approach requires holding out a portion of the training set for testing, which makes it data-inefficient with negative impact on the post-training performance. In view of these considerations, it would be highly desirable to develop a new general-purpose technique able to set the data free in their double role of (i) *delivering the information that is needed to learn*, while also (ii) *providing evaluations on the attained generalization result.*

In this paper, we move towards the goal expressed in (i) and (ii). To this aim, we build upon a recent breakthrough in the field of compression schemes, (Campi & Garatti, 2023). Therein, the authors have established new and tight bounds for the so-called probability of change of compression that hold under certain properties: under the property of *preference*, tight upper bounds have been obtained that link the probability of change of compression to the size of the compressed set[1]; moreover, under a so-called *non-associativity* property and a property of *non-concentrated mass*, a lower bound has also been obtained. Hence, under these additional properties, the change of compression is put in sandwich between two bounds, which are shown to rapidly converge one on top of the other as the number of data points increases. The interest of these results in the context of statistical learning lies on the fact that some learning algorithms naturally define a compression scheme and the probability of change of compression can be used to bound the probability of misclassification/misprediction. However, the challenge is that many other learning algorithms (including most algorithms for deep learning) do not exhibit any compression property.

With this paper, we give new, significant, thrust to the above findings. Specifically, we present a *meta-algorithm*, called Pick-to-Learn (P2L), that incorporates any learning algorithm given as a black box and *makes it* into a compression scheme, so licensing the use of results from compression to establish generalization bounds in virtually any learning problem. More precisely, the meta-algorithm constructs a loop around the original learning algorithm that elects at each iteration a new data point to be included in the set of those used for training. Independently of the inner algorithm (which can be any algorithm, e.g., GD – Gradient Descent – for regression), we show that the meta-algorithm possesses the properties of *preference*, *non-associativity* and *non-concentrated mass* by which, as mentioned above, one can secure extremely tight bounds on the change of compression. By further linking the change of compression to the probability of misclassification (or misprediction, in regression problems), powerful and informative generalization bounds are obtained without requiring the use of test-sets. When using our approach on various setups, of which we here present the MNIST classification problem and a synthetic regression problem, we find that P2L returns bounds that are comparable, or superior, to those attained via test-set and PAC-Bayes bounds, while it learns models with better post-training performance.

The idea and the theoretical apparatus behind P2L is presented for the first time in this paper, which also provides a complete set of proofs (given in the appendices) to establish the ensuing generalization results. On the other hand, it has to be said that the usage of the meta-algorithm P2L requires the user to choose a rule by which a new data point is selected at each iteration of the external loop, a choice that is problem-dependent and cannot be part of the general theory. Therefore, the present contribution lays the groundwork of a general new methodology and we expect that this methodology will thrive in the coming few years by adapting it to various specific contexts.

While our demonstrative focus in this paper is on classification and regression, P2L bears a promise of applicability to any data-driven learning problem, including large-scale constrained optimization. Interestingly, in recent years, selection rules to iteratively find "core" examples in the training dataset have been studied in various learning problems, see, e.g., (Toneva et al., 2018; Paul et al., 2021; Yang et al., 2022). These works differ from the present contribution both in their motivations (pruning the training dataset is motivated by computational issues) and results (none of these works provide a compression scheme according to the classical definition and certainly they do not enforce any preference property). Nonetheless, these selection rules can be adopted in the external loop of P2L

---

[1]The notion of preference was known in the machine learning literature before (Campi & Garatti, 2023) where it is often referred to as *stability*. Under stability, previous contributions, e.g., (Bousquet et al., 2020; Hanneke & Kontorovich, 2021), have provided valuable generalization results and the approach we propose may as well build upon these results; however, the bounds in (Campi & Garatti, 2023) largely improve over all previous findings.

and we envisage a synergy of our results with these methods. This opens exciting perspectives for future research.

**Structure of the paper.** After introducing the mathematical preliminaries (Section 2), we present the meta-algorithm P2L (Section 3) and its generalization results (Section 4). For the sake of generality, these sections adopt an *abstract viewpoint* that accommodates various learning frameworks. Applications to MNIST classification and synthetic regression can be found in Sections 5 and 6. Appendix A contains the proof of the main result and various additional results, including an extended version of P2L. Appendix B instead includes some implementation details and additional material for the MNIST application. Finally, Appendix C provides a further numerical experimentation to study the interpretability of the results of P2L. The source code necessary to reproduce all our numerical results can be found in the Supplementary Material, and at at `https://github.com/dario-p/P2L`.

## 2 Mathematical Preliminaries

We use $z$ to denote an example, which is an element from a generic set $\mathcal{Z}$. For instance, in supervised classification, $z$ is an input-label pair $(x, y)$ and $\mathcal{Z} = \mathcal{X} \times \{1, 2, \ldots, M\}$; instead, in supervised regression, $\mathcal{Z} = \mathcal{X} \times \mathbb{R}$. We assume to have access to $N$ examples, which we collect in a dataset $D$; that is, $D = \{z_1, z_2, \ldots, z_N\}$. Throughout, we model $D$ as a *multiset*.[2] This is motivated by the fact that P2L's output will not depend on the order of appearance of the examples but it will account for (possibly) repeated observations. With multisets, the set operations $\cup, \cap, \backslash, \subseteq$ extend from their definitions for sets in an obvious way.[3] All multisets encountered in our mathematical derivations have a finite number of elements and $|\cdot|$ denotes the cardinality where each element is counted as many times as is its multiplicity. The multiset of examples $D$ is modeled as a realization of $\boldsymbol{D} = \{\boldsymbol{z}_1, \boldsymbol{z}_2, \ldots, \boldsymbol{z}_N\}$, where $\boldsymbol{z}_1, \boldsymbol{z}_2, \ldots, \boldsymbol{z}_N$ are independent and identically distributed (i.i.d.) random elements taking value in $\mathcal{Z}$ and defined over a probability space $(\Omega, \mathcal{F}, \mathbb{P})$.[4]

We wish to utilize $D$ to construct a hypothesis from a hypothesis space $\mathcal{H}$, which can be any generic space. For example, if we are tasked with making predictions on the label/output of previously unseen inputs, $\mathcal{H}$ can be the space of classifiers/predictors obtained by suitably parameterized neural networks. We assume to be given a *learning algorithm $L$* that maps a *list* of examples to a hypothesis (this is the inner black-box of P2L). Unlike multisets, lists come with a positional order of their elements and our approach works rigorously with learning algorithms $L$ whose returned hypothesis either depends on this positional order or does not. Considering a list (as opposed to a multiset) as input to $L$ gives a setup that embraces the most general perspective. This is important as many commonly employed algorithms, e.g., SGD for the training of neural networks, are of the first type.

## 3 The Meta-Algorithm P2L

As anticipated, our goal is that of utilizing a given learning algorithm $L$ as a building block to construct a meta-algorithm (P2L) that induces a compression scheme with desired properties, regardless of whether the initial learning algorithm has such properties. For this, we need two ingredients: (i) an initial hypothesis $h_0 \in \mathcal{H}$; (ii) a suitable *criterion of appropriateness*, and a corresponding *appropriateness threshold*. The criterion of appropriateness quantifies to *what extent* a given hypothesis $h$ is appropriate for an example $z \in \mathcal{Z}$, while the threshold is used to assess – in the form of a yes/no answer – *whether* the given hypothesis $h$ is deemed sufficiently appropriate for $z$. We formalize the idea of criterion of appropriateness, and the corresponding threshold, by introducing a *hypothesis-dependent total order* $\leq_h$ over the extended set $\mathcal{Z}_{\mathbb{S}} = \mathcal{Z} \cup \{\texttt{Stop}\}$, where $\texttt{Stop}$ is an external element that is added to $\mathcal{Z}$ for algorithmic reasons. Given a hypothesis $h$ (our meta-algorithm iterates over subsequent choices of $h$ before exiting and $\leq_h$ will be used at each iteration with the current $h$), $\leq_h$ orders the elements in $\mathcal{Z}$ and, in particular, allows us to determine the element that

---

[2]This means that, e.g., $\{1, 1, 2\}$ is the same as $\{1, 2, 1\}$, but both are different from $\{1, 2\}$ because in a multiset, like in a set, there is no concept of order of the elements but account is taken of their multiplicity.

[3]This is easily achieved using the multiplicity function $\mu_U$ for a multiset $U$, which counts how many times each element of $\mathcal{Z}$ occurs in $U$. Then, $\mu_{U \cup U'}(z) = \mu_U(z) + \mu_{U'}(z)$, $\mu_{U \cap U'}(z) = \min\{\mu_U(z), \mu_{U'}(z)\}$, and $\mu_{U \backslash U'}(z) = \max\{0, \mu_U(z) - \mu_{U'}(z)\}$. Finally, $U \subseteq U'$ iff $\mu_U(z) \leq \mu_{U'}(z)$ for all $z$.

[4]Throughout, boldface denotes random quantities and we tacitly assume that all random quantities that we introduce are measurable.

is the least appropriate. The outside element `Stop` is instead used to model the appropriateness threshold: $h$ is "enough appropriate" for $z \in \mathcal{Z}$ if $z \leq_h$ `Stop` while it is not appropriate enough if `Stop` $\leq_h z$. Note that the two conditions cannot be satisfied simultaneously because $z \neq$ `Stop` (recall that `Stop` is an outside element) and $\leq_h$ is a total order. While remarking the generality of this setting, we provide two examples for concreteness and clarity of presentation.

**Example 3.1** (Classification). Consider a binary classification problem where $z = (x, y)$ with $x \in \mathcal{X}$ and $y \in \{0, 1\}$. Commonly employed hypotheses (e.g., neural networks) return the probability that a feature vector is mapped to either of the two classes. A possible criterion measuring the appropriateness of $h$ for $z$ is given by the cross-entropy between the label $y$ and the probability distribution returned by $h$ in correspondence of $x$, (Bishop & Nasrabadi, 2006)[Sec. 4.3.4]. Then, the appropriateness threshold can be used to certify if $z$ is correctly classified by $h$ by assessing if the cross-entropy is above or below the threshold $-\ln(0.5)$. This setting is captured by the following total order: for $z_1, z_2 \in \mathcal{Z}$, we define $z_1 \leq_h z_2$ if the cross-entropy of $z_1$ is no bigger than the cross-entropy of $z_2$ (and ties are broken according to any given rule), while `Stop` $\leq_h z$ if $z$ has cross-entropy larger than $-\ln(0.5)$ and $z \leq_h$ `Stop` otherwise.

**Example 3.2** (Regression). In regression, $z = (x, y)$ with $x \in \mathcal{X}$ and, for example, $y \in \mathbb{R}$. In this setting, a hypothesis is typically a predictor that maps $x$ into an estimate $\hat{y}(x)$ for $y$. A possible criterion measuring the appropriateness of $h$ for $z$ is given by $|y - \hat{y}(x)|$, while the appropriateness threshold can be specified by requiring that $|y - \hat{y}(x)| \leq \gamma$ for a given $\gamma$ for which the predictions are sufficiently informative. In this case, the hypothesis-dependent total order can be defined as follows. For $z_1, z_2 \in \mathcal{Z}$, $z_1 \leq_h z_2$ when $|y_1 - \hat{y}(x_1)| \leq |y_2 - \hat{y}(x_2)|$ (when $|y_1 - \hat{y}(x_1)| = |y_2 - \hat{y}(x_2)|$ but $z_1 \neq z_2$ the tie can be broken according to any given rule); otherwise, `Stop` $\leq_h z$ when $|y - \hat{y}(x)| > \gamma$ and $z \leq_h$ `Stop` when $|y - \hat{y}(x)| \leq \gamma$.

We are now ready to introduce the meta-algorithm P2L (Algorithm 1). P2L works by iteratively feeding the learning algorithm $L$ with a growing list of training examples taken from $D$, while terminating when the current hypothesis is deemed sufficiently appropriate (as assessed by the appropriateness threshold) for all the remaining examples that have not been used for training. If this is not the case, P2L appends to the current training examples the example among those not yet used for which the current hypothesis is least appropriate (according to the criterion of appropriateness). P2L returns both the multiset $T$ of the employed training examples and the final hypothesis $h$.

Formally, given $(L, h_0, \leq_h)$, the meta-algorithm P2L is a map from a dataset $D$ to two objects: an hypothesis $h \in \mathcal{H}$, and a multiset $T \subseteq D$. We denote this with $(h, T) = \mathcal{A}(D)$. P2L is composed of an initialization, and a main loop where three steps are performed. See Algorithm 1, where $D_{\mathsf{S}} = D \cup \{\text{Stop}\}$ and $[T]_{\mathcal{A}}$ is the list of the elements in $T$ with positional order of each element corresponding to the iteration in which that element is selected by P2L.

---
**Algorithm 1** $\mathcal{A}(D)$ – The meta-algorithm P2L

---
1: **Initialize:** $T = \emptyset$, $h = h_0$, $\bar{z} = \max_{h_0}(D_{\mathsf{S}})$
2: **while** $\bar{z} \neq$ `Stop` **do**
3:     $T \leftarrow T \cup \{\bar{z}\}$         ▷ Augment $T$
4:     $h \leftarrow L([T]_{\mathcal{A}})$      ▷ Learn hypothesis
5:     $\bar{z} \leftarrow \max_h(D_{\mathsf{S}} \setminus T)$     ▷ Compute max
6: **end while**
7: **return** $h, T$    ▷ Hypothesis $h$ and multiset $T$

---

In the initialization, we let the multiset of training examples be empty, the hypothesis $h$ be $h_0$, and compute the maximal element in $D_{\mathsf{S}} = D \cup \{\text{Stop}\}$ according to $\leq_{h_0}$, which we denote with $\bar{z} = \max_{h_0}(D_{\mathsf{S}})$.[5] In the main loop, we first check if the maximal element is `Stop`. If so, the meta-algorithm terminates (thus our naming it `Stop`). Else, the training multiset is augmented with $\bar{z}$. We then produce a new hypothesis by running the learning algorithm on $[T]_{\mathcal{A}}$, i.e., we set $h \leftarrow L([T]_{\mathcal{A}})$. Finally, we compute the maximal element in $D_{\mathsf{S}} \setminus T$ according to the new hypothesis $h$, that is, $\bar{z} \leftarrow \max_h(D_{\mathsf{S}} \setminus T)$, and repeat. When the meta-algorithm terminates, it returns both the hypothesis $h$, and the multiset $T$.

We conclude with two important remarks. First, note that the choice of the learning algorithm $L$, ordering $\leq_h$, and initial hypothesis $h_0$ are arbitrary. Interestingly, strong generalization guarantees can be secured at this high level of abstraction (Theorem 4.2) and this allows one to tackle multiple learning problems, including both classification and regression (Sections 5 and 6). Second, observe that the generalization guarantees we will provide rely crucially on the fact that the meta-algorithm

---
[5]Recall that the maximal element is unique.

P2L possesses a key feature: P2L *compresses* the multiset $D$ into the multiset $T$, and applying P2L to $D$ or to $T$ reconstructs the same output.

# 4 Statistical Risk and Main Result

In this section we provide formal generalization guarantees on the performance of the hypothesis obtained by running the meta-algorithm P2L. Towards this goal, we introduce the notion of statistical risk, which we will then bound.

**Definition 4.1** (Statistical Risk). The statistical risk of a given hypothesis $h \in \mathcal{H}$ is $R(h) = \mathbb{P}\{\texttt{Stop} \leq_h z\}$, where $z$ is a random example independent of and distributed as each $z_i$.

Informally, the statistical risk of a hypothesis $h$ measures the probability that $h$ is not appropriate for a new, out-of-sample, example $z$. Given the generality of $\leq_h$, this notion can embody various specifications. For instance, in the context of Example 3.1 the statistical risk coincides with the probability of misclassification; in the context of Example 3.2 with the probability of misprediction above a level $\gamma$.

In the result stated below, $\boldsymbol{D}$ is seen as a random element (with an unknown probability distribution) so as to capture the variability in the dataset. Correspondingly, $\boldsymbol{h}$ is a random element and the risk $R(\boldsymbol{h})$ is a random variable. Our objective is to show that $R(\boldsymbol{h})$ can be assessed based on the compression level achieved by P2L, i.e., on the cardinality of $\boldsymbol{T}$. We aim to make a statement of the form "With high probability $1 - \delta$ with respect to the $N$ i.i.d. draws generating the dataset (i.e, with respect to the unknown probability distribution of $\boldsymbol{D}$), the statistical risk is upper bounded by $\overline{\varepsilon}(|\boldsymbol{T}|, \delta)$", for a suitable choice of $\overline{\varepsilon}(\cdot, \cdot)$. Towards defining $\overline{\varepsilon}(\cdot, \cdot)$, we need to introduce the following function $\Psi_{k,\delta} : (-\infty, 1) \to \mathbb{R}$ indexed by $k = 0, 1, \ldots, N$ and by the confidence parameter $\delta \in (0, 1)$:

$$\Psi_{k,\delta}(\varepsilon) = \frac{\delta}{2N} \sum_{m=k}^{N-1} \frac{\binom{m}{k}}{\binom{N}{k}} (1-\varepsilon)^{-(N-m)} + \frac{\delta}{6N} \sum_{m=N+1}^{4N} \frac{\binom{m}{k}}{\binom{N}{k}} (1-\varepsilon)^{m-N},$$

where the first summand evaluates to zero when $k = N$. We are now ready to state our main result.

**Theorem 4.2** (Bound on Statistical Risk). *Let $(\boldsymbol{h}, \boldsymbol{T}) = \mathcal{A}(\boldsymbol{D})$, the output of P2L. For any $\delta \in (0, 1)$ it holds that*

$$\mathbb{P}\{R(\boldsymbol{h}) \leq \overline{\varepsilon}(|\boldsymbol{T}|, \delta)\} \geq 1 - \delta, \tag{1}$$

*where, for $k = 0, 1, \ldots, N - 1$, $\overline{\varepsilon}(k, \delta)$ is the unique solution to the equation $\Psi_{k,\delta}(\varepsilon) = 1$ in the interval $[k/N, 1]$, while $\overline{\varepsilon}(N, \delta) = 1$.*

See Appendix A for a proof. Theorem 4.2 unveils a deep connection between the level of compression achieved by $\mathcal{A}$, as measured by the cardinality $|T|$, and the generalization level of the hypothesis. Quantitatively, the link is dictated by $\overline{\varepsilon}(k, \delta)$, whose trend is exemplified in Figure 1 for different values of $\delta$.

Three important observations are in order. First, higher compression, i.e., a lower $|T|$, corresponds to better generalization guarantees. Second, the dependence of $\overline{\varepsilon}(k, \delta)$ on $\delta$ is provably logarithmic. Hence, selecting a very small value of $\delta$, say $10^{-6}$, i.e., asking for $R(\boldsymbol{h}) \leq \overline{\varepsilon}(|\boldsymbol{T}|, \delta)$ to hold with high confidence, say 0.999999, in practice *with certainty*, has little cost. Third, we notice that the bound can be applied without knowledge of the probability distribution by which data are generated (*distribution-free* result).

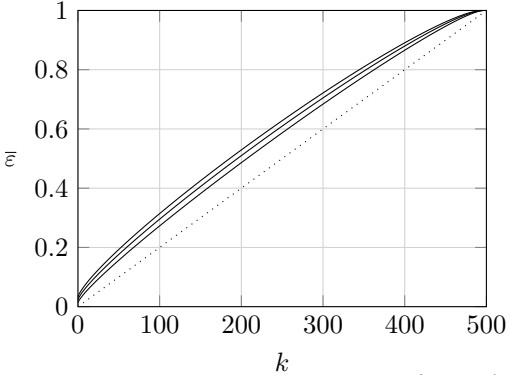

Figure 1: Function $\overline{\varepsilon}(k, \delta)$, $\delta = 10^{-6}, 10^{-4}, 10^{-2}$, (top to bottom). $N = 500$. The dotted line is $k/N$.

# 5 Application to MNIST Classification

In this section we apply P2L to the MNIST digit recognition problem. Our goal is to compare the post-training performance and generalization bounds obtained with an implementation of P2L, PAC-Bayes and test-set approaches. The latter two are known to give the best generalization bounds to date.

More precisely, we consider a binary version of the MNIST dataset introduced in the seminal work of (Dziugaite & Roy, 2017) for the specific purpose of comparing the generalization guarantees of PAC-Bayes and other approaches, and later employed in, e.g., (Rivasplata et al., 2019). In this problem, the digits 0-4 and 5-9 are mapped to the labels 0 and 1. To classify the inputs, we employ a fully connected feed-forward neural network with three hidden layers each with 600 nodes and ReLu activation functions. The input has 784 nodes and the output has two nodes which are passed to a softmax function. This architecture, employed in the above-cited works, is used across all our experiments.

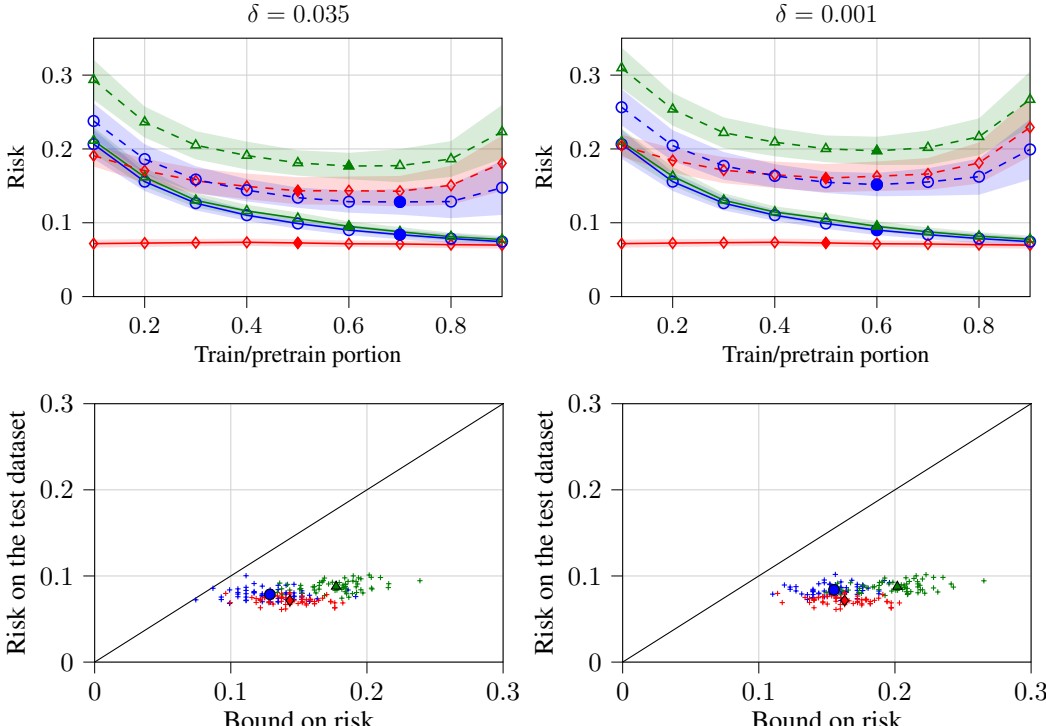

Figure 2: Top row: Average bounds on the risk (dashed) and misclassification on the test dataset (solid) ± one standard deviation for P2L ◊, test-set ○, and PAC-Bayes △ with a confidence of $\delta = 0.035$ (left) and $\delta = 0.001$ (right). The solid markers denote the *best bounds* achieved and their corresponding post-training performance. Bottom row: Distribution of the upper bounds on the risk and misclassification on the test dataset for the models achieving the *best bounds* for P2L +, test-set +, and PAC-Bayes +. Precisely, each point in these figures corresponds to the pair (*generalization bound*, *actual misclassification level*) achieved in one of the 60 partitions of the MNIST training dataset we use. The means are indicated with a solid diamond, circle, and triangle, respectively.

MNIST consists of a training dataset containing 60000 examples and a test dataset with 10000 examples. In our experiments, we train all models only on 1000 samples from the training dataset at a time. We particularly care of this "small" dataset setting, in which one has to make the most of the data for both training and assessing the generalization. This is a setting of interest to various fields where data are a limited, possibly costly, resource.[6] Specifically, we shuffle the original MNIST training dataset and extract 60 disjoints datasets with 1000 data points each. All approaches we compare are run on the resulting 60 datasets, which are used to both train the network and provide generalization bounds. The *full* test dataset, containing 10000 examples, is never used in any training phase. It is instead used to evaluate the actual post-training performance of each trained model. We

---

[6]Our experience ranges from applications to cardiac defibrillation in which patients in out-of-hospital cardiac arrest are classified as being able or not able to positively react to a defibrillation shock (in the latter, alternatives are possible, chiefly administrating a cardiac massage prior to the shock), to applications with spacesuit testing in which suits are artificially bombarded with particles to mimic the condition encountered in the outer space. While in the first example the number of recorded cases is potentially very high (modern defibrillators automatically collect ECG traces), still these data are not made public and easy to secure in large quantities, while in the second example each single data point can be quite costly, suggesting that the dataset has to be kept as small as possible.

now present the specific algorithmic choices made for P2L. Due to space limitations, the details concerning the test-set approach and PAC-Bayes can be found in Appendix B.1.

**P2L.** As discussed, P2L is fully specified once a learning algorithm $L$, an initial hypothesis $h_0$, and a hypothesis-dependent total ordering $\leq_h$ with a criterion of stop are defined. For the learning algorithm $L$, we choose the widely employed GD (gradient descent) with momentum, which we run for 200 epochs with a learning rate of $0.01$, momentum of $0.95$, dropout probability of $0.2$. For the sake of clarity, we remark that each time GD is called by P2L, GD iterates until convergence (or until reaching the allowed number of epochs) so as to perform the minimization of the empirical risk. As initialization of GD, we take the $h$ returned at the previous iteration of P2L.[7] Regarding $h_0$, it is clear that when starting from an educated guess the choice of the worst example to be inserted in $T$ tends to be more meaningful, allowing P2L to terminate with a smaller $T$ and, thus, a better generalization bound. To this purpose, one can use a portion of each training dataset to pretrain $h_0$ (using GD again), while P2L and, therefore, the ensuing generalization bound rely on the remaining part of the training dataset. In our computation, we experiment with multiple sizes of the pretraining portion. As for $\leq_h$, given an hypothesis $h$, i.e., a neural network, we use the total order induced by the cross-entropy of its output as discussed in Example 3.1 (ties are broken according to the lexicographic order of the examples' bit representation). This choice ensures that the statistical risk of Definition 4.1 corresponds to the probability of misclassification, and thus a bound on the statistical risk coincides with a bound on the misclassification. The final bound we present on the misclassification is that of the main result in Theorem 4.2, where the number of samples equals $N$ minus those used to pretrain the initial hypothesis $h_0$.

**Experimental results.** The results are presented in Figure 2 for different values of the confidence. The top row depicts, the average (over the 60 trials) generalization bounds as well as the average misclassification levels on the test dataset (post-training performances) with their dispersion for the three approaches. The values are plotted as functions of the portion of the dataset (called train/pretrain portion) used to train the the model in GD+test-set, to pretrain the prior distribution in PAC-Bayes, and to pretrain the initial hypothesis $h_0$ in P2L. The joint distribution of the returned bound and actual post-training misclassification level for the best models (corresponding to train/pretrain portion equal to 0.5, 0.6, 0.7 for P2L, PAC-Bayes, and test-set approach, respectively) are shown in the bottom row. The average bounds and misclassification levels for these best models are also presented in Table 1 for $\delta = 0.035$. For the sake of completeness, Table 1 also reports the average running times of one execution (i.e., for one dataset with 1000 examples) of the three approaches (computational resource: Apple MacBook Pro with M1 Pro CPU and 32Gb of ram).

Table 1: Risk of *best models* for $\delta = 0.035$.

|  | Bound on risk | Risk on the test dataset | Difference | Average running time |
|---|---|---|---|---|
| P2L | 0.143 | 0.072 | 0.071 | 2m 1s |
| Test-set | 0.129 | 0.079 | 0.050 | 0m 5s |
| Pac-Bayes | 0.177 | 0.088 | 0.089 | 4m 1s |

**Conclusions.** Four important observations are in order. First, it is evident that, in the present application, the implementation of P2L with GD outperforms the PAC-Bayes approach with respect to both the provided upper bound on the risk and the post-training performance on the test dataset. This is true not just for the best model learned with each approach (cfr. the solid diamonds and the solid triangles in Figure 2), but it holds uniformly across all train/pretrain portions we tried (cfr. the green and red curves). Notably, the model returned by P2L+GD when using only 10% of the data to pretrain the initial hypothesis $h_0$ has a risk bound that is comparable to that of the *best* PAC-Bayes model (whose prior is trained on 60% of the data) while it achieves a better post-training performance. Second, P2L+GD provides best risk bounds that are comparable to those of GD+test-set approach, albeit slightly inferior (0.143 vs 0.129 for $\delta = 0.035$ and 0.163 vs 0.155 for $\delta = 0.001$). However, P2L+GD provides better post-training performance uniformly across all train/pretrain portions. As a matter of fact, for any choice of pretrain portion, P2L+GD's performance is equal to that obtained when GD is run on the whole data set ($N = 1000$), for which the test-set approach cannot provide

---

[7]As an alternative, one could also re-train a model from scratch at every iteration. This implementation of GD can be seen as a slightly different choice of the inner algorithm $L$ for P2L. The selected implementation of GD is motivated by the ensuing computational advantage, since fewer gradient steps are typically needed to converge, and the fact that this implementation makes $h$ less sensitive to the addition of one example in $T$, which is helpful in later stages of P2L to speed up its termination.

any meaningful bound since no data are left for testing the model. This is a clear indication of the important feature that *P2L utilizes all data to jointly learn a good model and provide a risk bound.* Third, the post-training performance of PAC-Bayes and GD+test-set approaches are similar. This suggests that in PAC-Bayes the training of the posterior does not exploit the additional available data to improve the model, but rather to certify it, in a similar vein as in the test-set approach. This fact has also been observed recently in (Lotfi et al., 2022)[Fig 1(a)]. Fourth, as for the tightness of the bounds (i.e., the difference between the upper bounds and risk on the test dataset), GD+test-set approach provides the tightest results, while P2L+GD and PAC-Bayes approaches alternate depending on the specific train/pretrain portion selected.

## 6   Application to a synthetic regression problem

In this section we apply our methodology to a synthetic regression problem in order to showcase the flexibility of the approach introduced in Section 4.

Specifically, we consider 100 distinct training datasets with $N = 200$ examples $(x, y)$ and then another 100 distinct training datasets with $N = 500$ examples $(x, y)$. The input $x \in [-0.5, 1.5]$ is extracted uniformly at random and the output is $y = f(x) + e$ with $f(x) = \sin(2.5\pi x)/(2.5\pi x)$ and $e$ extracted according to a normal distribution with zero mean and standard deviation $0.05$, see Figure 3. For each dataset, our objective is twofold: finding a model that fits the data well, and providing a bound on the probability that the model mispredicts the output by more than a threshold $\gamma = 0.1$ (bounds of this sort matters in relation to many applications).

In pursuing this goal, we compare the generalization bounds and post-training performances attained using P2L and test-set bounds. For illustration purposes, in all our experiments we consider a simple network architecture comprising one input node, one output node, and one hidden layer with six nodes, each equipped with a $\tanh$ activation function. To evaluate the post-training performance, we use an additional test dataset with 20000 examples. The value of $\delta$ is set to $0.035$.

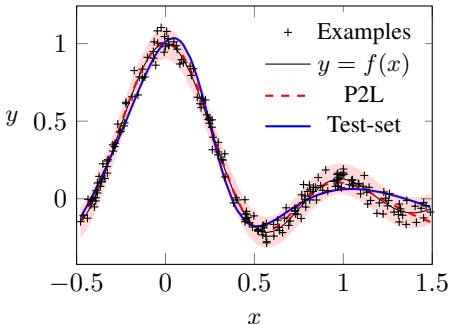

Figure 3: An instance of dataset used for regression. The crosses are examples $(x, y)$. The black line depicts the function $f(x) = \sin(2.5\pi x)/(2.5\pi x)$ around which noise is added. The red dashed and blue solid lines represent the fit obtained with a choice of train/pretrain portion equal $0.3$ (best) for P2L and test-set approaches. The red shaded region depicts a tube of radius $0.1$ around the red model. The risk of the red model is the probability that an unseen example falls outside the shaded region.

**P2L.** In deploying P2L we use GD as learning algorithm $L$, and, similarly to the MNIST example, we experiment with initial hypotheses $h_0$ trained on different fractions of the training dataset (including the null fraction in which case $h_0$ is the network with all weights set to 0). We fix the same total order $\leq_h$ of Example 3.2, with $\gamma = 0.1$. In this context, the risk introduced in Definition 4.1 measures the probability that the output $y$ is mispredicted by more than $\gamma$, thus giving us a guarantee on the quality of the model. The generalization bound we use is that in Theorem 4.2, where the number of samples equals $N$ minus those used to pretrain the initial hypothesis $h_0$. We use a learning rate of $0.1$, momentum of $0.95$, 1000 epochs, and no dropout.

**Test-set approach.** We apportion each dataset in two, with one portion used for training, and the other for deriving the generalization bounds. We run GD for 1000 epochs, perform a grid search over learning rates [0.001, 0.005, 0.01], momentum [0.9, 0.95] (no dropout), and select those giving the best generalization bound. As in the MNIST example (see Appendix B.1), we use the binomial test-set bound of (Langford, 2005)[Thm 3.3] with a number of samples equal to $N$ minus those used for training.

**Experimental results.** The first two panels of Figure 4 present the average (over the 100 trials) upper bounds on the risk and the average risk on the test dataset jointly with their dispersion, as a function of the data portion used to train the model (GD+test-set) or to pretrain the initial hypothesis (P2L). The third and fourth panels depict the distribution of the upper bound and of the risk on the test dataset

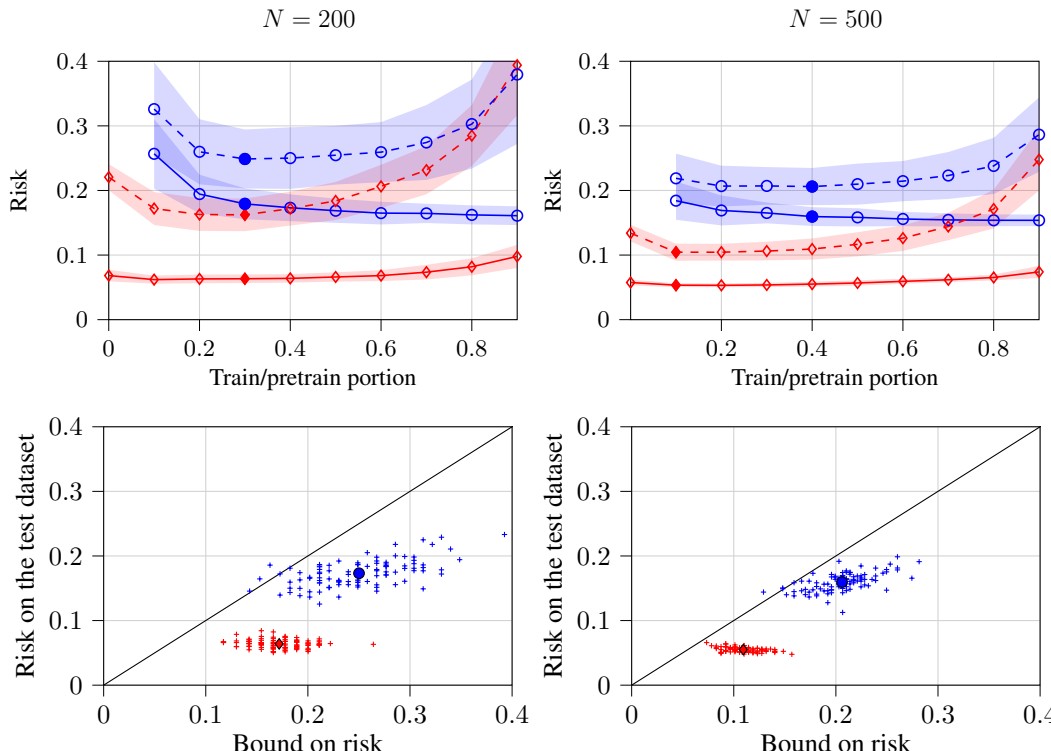

Figure 4: Top row: Average bounds on the risk (dashed) and risk on the test dataset (solid) ± one standard deviation for P2L ◇ and test-set ○ approaches, with $N = 200$ (left) and $N = 500$ (right). The solid markers denote the best average bounds and the corresponding risks on the test dataset. Bottom row: Empirical distribution of the bound and risk on the test dataset for the data split returning the best average bound for P2L + and test-set +. Their means are indicated with a solid diamond and circle, respectively. Note that the test-set approach is not considered for zero train portion, as it always requires a non-zero amount of data to train the model. Observe that three instances in the test-set case (out of one hundred) in the bottom row violate the upper bound. This is in line with the choice of $\delta = 0.035$, suggesting that the bound will fail, on average, on 3.5 datasets out of one hundred.

for the data proportion 0.3, returning the best learned models for both P2L and the test-set approach. Their averages are compared in Table 2, which also includes the average runtime (computational resource: Apple MacBook Pro with M1 Pro CPU and 32Gb of ram).

Table 2: Risk of *best models* for $N = 200$, $\delta = 0.035$

|          | Bound on risk | Risk on the test dataset | Difference | Average running time |
|----------|---------------|--------------------------|------------|----------------------|
| P2L      | 0.172         | 0.064                    | 0.108      | 1.18s                |
| Test-set | 0.250         | 0.173                    | 0.077      | 0.21s                |

**Conclusions.** First, it appears evident that, in this synthetic regression problem, the application of P2L with GD provides superior results to those achieved by GD with the test-set approach, regardless of the size of the datasets used for training. This holds true jointly for the upper bound and the risk on the test dataset uniformly across all train/pretrain portions (cfr. the red and blue dashed lines, similarly for the solid ones). We ascribe this result to the fact that P2L does not set aside data for testing, and yet it also provides rigorous evaluations of the risk. Second, the test-set approach provides bounds that are closer to the risk it incurs on the test dataset for $N = 500$, while this effect is less clear with $N = 200$. Third, as expected, when the size of the training dataset grows, both the upper bounds and the risk on the test dataset improve.

To conclude, it is fair to notice that GD does not pursue directly the goal of obtaining a prediction error smaller than $\gamma$, and this may justify the large gap in the post-training performance between

the two approaches also when almost all the dataset is used for training in the test-set approach. Although GD is used as inner algorithm $L$ in P2L, it seems that a good performance with respect to the chosen appropriateness criterion can be obtained thanks to the structure of the meta-algorithm. This is another interesting feature of P2L.

# 7    Conclusions, Limitations and Future research

We have proposed a novel framework called P2L to provide virtually any learning algorithm with sharp generalization bounds. Our approach is based on making a given learning algorithm into a compression scheme with desirable properties, thus enabling the use of powerful generalization results. Numerical results show that P2L is capable of learning hypotheses with post-training performances and generalization bounds equal or superior to the state of the art.

**Computational aspects.** While P2L requires learning a hypothesis over a training set of increasing size, and thus might be less efficient than learning the hypothesis only once over the full dataset, it is important to look at it from the appropriate perspective: we are concerned with settings where data is limited or costly to acquire (see, e.g., Footnote 6 and ensuing discussion) while computations are performed off-line and fast execution does not represent a primary concern (a setup also considered in other recent work, e.g., (Foong et al., 2021)).

**Source of conservatism.** In our framework, as revealed by the proof of Theorem 4.2, the risk is controlled by bounding the probability of change of compression (as defined in Equation (2) in Appendix A.1). Notably, the mismatch between the risk and the probability of change of compression is the *only* source of conservatism our approach needs to resolve since the upper and lower bounds on the probability of change of compression from Theorem A.4 are *extremely tight* – see Remark A.9. The magnitude of this conservatism is determined by the choices we make in specializing the proposed meta-algorithm. For a given learning algorithm $L$, these choices entail selecting an initial hypothesis $h_0$ and an hypothesis-dependent total order used to select which data points are fed to the learning algorithm $L$. Our experimenting with multiple initial hypotheses (trained with different portions of the dataset) was *solely* geared at reducing this gap. Indeed, after training $h_0$, P2L allows the data to "freely speak", and thus improve the resulting hypothesis, as it can be appreciated from the fact that the misclassification on the test dataset for P2L is constant across *all* prior/train portions (see Figure 2, top row). This is in stark contrast with the test-set approach and even PAC-Bayes. In the former, data are either used to train the model *or* to provide a risk bound. In the latter, data employed to train the posterior are effectively used to compute a risk bound as opposed to significantly improving the quality of the prior (see first row in Figure 2). We conclude noting that our choices of the initial hypothesis and total order is but one of many possible. Depending on the specific learning problem, other choices can be made (see for examples the last part of the Introduction). We believe that the overall fact that the theoretical apparatus in our approach clearly identifies the sole source of conservatism will put us and others in the position to build upon the P2L framework beyond this work.

# Acknowledgments

This paper was partly supported by FAIR (Future Artificial Intelligence Research) project, funded by the NextGenerationEU program within the PNRR-PE-AI scheme (M4C2, Investment 1.3, Line on Artificial Intelligence); and the EPSRC grant EP/Y001001/1, funded by the International Science Partnerships Fund (ISPF) and UKRI.

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

# Appendix

## A  Proof of Theorem 4.2 and Additional Results

In this section we prove Theorem 4.2. En route to this goal, we will also present additional results that help better position our contribution. Doing so will allow us to highlight directions where we believe there is room for exciting future work. The section is conceptually divided in three portions: in Appendix A.1 we recall a relevant result from (Campi & Garatti, 2023) along with the necessary mathematical background; this result is then leveraged in Appendix A.2 to prove Theorem 4.2 and other related results; finally, in Appendix A.4 we revisit the P2L framework in the light of some technical details revealed by the proof of Theorem 4.2.

### A.1  Preferent Compression Gives Tight Risk Bounds

We begin with introducing two properties of a compression function $\mathsf{c}$ (*preference* and *non-associativity*) and the property of *non-concentrated mass* from which strong generalization guarantees follow. Recall that a compression function $\mathsf{c}$ is a map from any multiset $D$ of elements in $\mathcal{Z}$ to a sub-multiset $\mathsf{c}(D) \subseteq D$.[8] Throughout, we use $\mathsf{c}(D, z)$ as a shorthand for $\mathsf{c}(D \cup \{z\})$, and likewise for $\mathsf{c}(D, z_1, \dots, z_p)$. Moreover, $\boldsymbol{z}$ denotes a random example independent of and identically distributed as each $\boldsymbol{z}_i$.

*Property* A.1 (Preference). For any pair of multisets $V$ and $D$ of elements of $\mathcal{Z}$ with $V \subseteq D$, $\mathsf{c}(D) \neq V \implies \mathsf{c}(D, z) \neq V, \ \forall z \in \mathcal{Z}$.

The *preference* property asserts that, if the multiset $V$ was not chosen when compressing $D$, it will also not be chosen when compressing the set $D$ augmented with one more element $z$. Under the preference property, it is easy to prove that $\mathsf{c}(\mathsf{c}(D)) = \mathsf{c}(D)$, i.e., the compression of a compressed multiset must be the compressed multiset itself.

*Property* A.2 (Non-associativity). For any multiset $D$ of elements of $\mathcal{Z}$ and for any $z_1, \dots, z_p \in \mathcal{Z}$, $p > 1$, $\mathsf{c}(D, z_i) = \mathsf{c}(D), \ \forall i \in \{1, \dots, p\} \implies \mathsf{c}(D, z_1 \dots, z_p) = \mathsf{c}(D)$.

The *non-associativity* property ensures that, if the compression does not change by adding one element at a time, it must also not change when adding all the elements together.[9]

*Property* A.3 (Non-concentrated mass). $\mathbb{P}\{\boldsymbol{z} = z\} = 0, \forall z \in \mathcal{Z}$.

The property of *non-concentrated mass* ensures that every point $z \in \mathcal{Z}$ is drawn with probability zero.

Under these three properties, strong results on the probability of change of compression, formally defined as

$$\phi(\boldsymbol{D}) = \mathbb{P}\{\mathsf{c}(\mathsf{c}(\boldsymbol{D}), \boldsymbol{z}) \neq \mathsf{c}(\boldsymbol{D}) \mid \boldsymbol{D}\} \tag{2}$$

(i.e., change of compression occurs when the compression of $\mathsf{c}(\boldsymbol{D})$ augmented with one element is different from $\mathsf{c}(\boldsymbol{D})$) are established in (Campi & Garatti, 2023).

**Theorem A.4** (Thm. 7 in (Campi & Garatti, 2023)). *Assume Properties A.1 to A.3. Then, for any $\delta \in (0, 1)$,*

$$\mathbb{P}\{\ \underline{\varepsilon}(|\mathsf{c}(\boldsymbol{D})|, \delta) \leq \phi(\boldsymbol{D}) \leq \overline{\varepsilon}(|\mathsf{c}(\boldsymbol{D})|, \delta)\ \} \geq 1 - \delta, \tag{3}$$

*where $\overline{\varepsilon}(k, \delta)$ for $k = 0, 1, \dots, N-1$ is the unique solution to $\Psi_{k,\delta}(\varepsilon) = 1$ in the interval $[k/N, 1]$, while $\overline{\varepsilon}(N, \delta) = 1$, and $\underline{\varepsilon}(k, \delta)$ for $k = 0, 1, \dots, N$ is the maximum between $0$ and the unique solution to $\Psi_{k,\delta}(\varepsilon) = 1$ in the interval $(-\infty, k/N]$ (function $\Psi_{k,\delta}(\varepsilon)$ is defined in Section 4).*

The previous statement asserts that, with probability $1 - \delta$ over the draws of $\boldsymbol{D}$, the probability of change of compression is contained in an interval with upper and lower extremes given by $\overline{\varepsilon}(k, \delta)$

---

[8]Note that this definition, and the properties that we shall introduce later in the section, apply not only to the multiset of actually observed examples, they apply to arbitrary multisets of any cardinality. Although we have used $D$ to avoid proliferation of the symbols, $D$ is used here, and elsewhere when needed, to denote generic multisets.

[9]Theorem A.4 continues to hold if the non-associativity property is required to apply $\mathbb{P}$-almost surely (as opposed to realization-wise). We use a realization-wise version for the sake of simplicity and also because this formulation suffices for the goals of this paper.

and $\underline{\varepsilon}(k, \delta)$, where $k$ is evaluated at the cardinality of the compression of the draw. Prompted by one of the reviewers, we gladly provide a concise outline of the proof structure for Theorem A.4 in the following Remark A.5.

*Remark* A.5 (Gist of the proof of Theorem A.4). *The proof of Theorem A.4 in (Campi & Garatti, 2023) starts by associating any compression scheme that satisfies Properties A.1 to A.3 to a probability measure from which one can compute $\mathbb{P}\{\underline{\varepsilon}(|\mathtt{c}(\boldsymbol{D})|, \delta) \leq \phi(\boldsymbol{D}) \leq \overline{\varepsilon}(|\mathtt{c}(\boldsymbol{D})|, \delta)\}$ and then by characterizing the class of all such probability measures in terms of certain conditions. The next step consists in obtaining an upper bound to $\mathbb{P}\{\underline{\varepsilon}(|\mathtt{c}(\boldsymbol{D})|, \delta) \leq \phi(\boldsymbol{D}) \leq \overline{\varepsilon}(|\mathtt{c}(\boldsymbol{D})|, \delta)\}$ by maximizing this quantity over the class of probability measures that has been previously characterized. The ensuing maximization problem is infinite dimensional and its solution is obtained by duality. Interestingly, no conservatism is introduced at this stage because strong duality holds. See (Campi & Garatti, 2023) for details.*

The interest of Theorem A.4 in the context of statistical learning lies in the fact that, whenever the compression function $\mathtt{c}$ ties in with the learning algorithm so that inappropriateness (e.g., misclassification or misprediction) implies change of compression, then the probability of inappropriate examples is dominated by the probability of change of compression and, therefore, it can be upper bounded with high confidence based on (3). See also Section 4 in (Campi & Garatti, 2023). This means that we can prove Theorem 4.2 by exhibiting a compression function that enjoys the result of Theorem A.4 and for which whenever $h(D)$ is inappropriate for $z$, i.e., $\mathtt{Stop} \leq_{h(D)} z$, we also have $\mathtt{c}(\mathtt{c}(D), z) \neq \mathtt{c}(D)$. This is indeed the demonstration path pursued in the next Appendix A.2.

## A.2 Proof of main result

Note that Theorem 4.2 does not assume the non-concentrated mass Property A.3. To ease the presentation, however, Theorem 4.2 will be first proven by assuming Property A.3, which allows for a more direct application of Theorem A.4; later, we will show how this extra assumption can be removed.

Suppose thus for the time being that Property A.3 holds true. Throughout this section, we denote with $\mathtt{c}_{\mathcal{A}}$ the compression function obtained by $\mathcal{A}$ when it compresses multiset $D$ into $T \subseteq D$ (recall that $(h, T) = \mathcal{A}(D)$). We show that $\mathtt{c}_{\mathcal{A}}$ enjoys the *preference* and *non-associativity* Properties A.1 and A.2. This licenses the use of Theorem A.4 to obtain upper and lower bounds on the probability of change of compression. We shall then relate these bounds on the probability of change of compression to the statistical risk as in Definition 4.1, so closing the proof when Property A.3 is assumed. We shall use $\mathcal{A}(D, z)$ and $\mathtt{c}_{\mathcal{A}}(D, z)$ as shorthand for $\mathcal{A}(D \cup \{z\})$ and $\mathtt{c}_{\mathcal{A}}(D \cup \{z\})$.

We begin by showing that $\mathtt{c}_{\mathcal{A}}(\cdot)$ is preferent.

**Lemma A.6.** *The compression function $\mathtt{c}_{\mathcal{A}}$ satisfies the preference Property A.1.*

*Proof.* Consider a multiset $D$ of elements of $\mathcal{Z}$ and further let $z \in \mathcal{Z}$. Suppose that $\mathtt{c}_{\mathcal{A}}(D, z) \neq \mathtt{c}_{\mathcal{A}}(D)$. Our goal is to show that in this case $\mathtt{c}_{\mathcal{A}}(D, z) \neq V$ for any $V \subseteq D$, from which Property A.1 immediately follows.
Denote by $T_k, h_k$ and $T_k', h_k'$ the multisets $T$ and hypotheses $h$ constructed in $\mathcal{A}(D)$ and $\mathcal{A}(D, z)$, respectively, at iteration $k$ of the "while" loop. $T_0 = \emptyset = T_0'$ and $h_0 = h_0'$ instead denote the empty multisets $T$ and initial hypotheses in the initialization step. Finally, let $D_{\mathtt{S}}' = D_{\mathtt{S}} \cup \{z\}$ (where we recall that $D_{\mathtt{S}} = D \cup \{\mathtt{Stop}\}$).
Since $\mathtt{c}_{\mathcal{A}}(D, z) \neq \mathtt{c}_{\mathcal{A}}(D)$, there must exist a $\bar{k} \geq 0$ such that the executions of $\mathcal{A}(D)$ and $\mathcal{A}(D, z)$ match for $k < \bar{k}$ (which implies that $T_k' = T_k$ and $h_k' = h_k$ for $k \leq \bar{k}$), but they differ at iteration $\bar{k}$ because, for $k = \bar{k}$, the maximal element computed in $\mathcal{A}(D)$ (line 5 if $\bar{k} \geq 1$ or line 1 if $\bar{k} = 0$) is different from the one computed in $\mathcal{A}(D, z)$. That is, $\max_{h_{\bar{k}}}(D_{\mathtt{S}} \setminus T_{\bar{k}}) \neq \max_{h_{\bar{k}}'}(D_{\mathtt{S}}' \setminus T_{\bar{k}}') = \max_{h_{\bar{k}}}(D_{\mathtt{S}}' \setminus T_{\bar{k}})$, where the last equality holds because, as previously noticed, $T_{\bar{k}}' = T_{\bar{k}}$ and $h_{\bar{k}}' = h_{\bar{k}}$.
We claim that it must be that $\max_{h_{\bar{k}}}(D_{\mathtt{S}}' \setminus T_{\bar{k}}) = z$ and $z \notin D_{\mathtt{S}} \setminus T_{\bar{k}}$. Indeed, if this was not the case, it would be that $\max_{h_{\bar{k}}}(D_{\mathtt{S}}' \setminus T_{\bar{k}}) \in D_{\mathtt{S}} \setminus T_{\bar{k}}$, which would imply that $\max_{h_{\bar{k}}}(D_{\mathtt{S}}' \setminus T_{\bar{k}}) = \max_{h_{\bar{k}}}(D_{\mathtt{S}} \setminus T_{\bar{k}})$ since $D_{\mathtt{S}} \setminus T_{\bar{k}} \subseteq D_{\mathtt{S}}' \setminus T_{\bar{k}}$. The latter would contradict $\max_{h_{\bar{k}}}(D_{\mathtt{S}} \setminus T_{\bar{k}}) \neq \max_{h_{\bar{k}}}(D_{\mathtt{S}}' \setminus T_{\bar{k}})$.
Since $\max_{h_{\bar{k}}}(D_{\mathtt{S}}' \setminus T_{\bar{k}}) = z \neq \mathtt{Stop}$, $\mathcal{A}(D, z)$ executes at least one more iteration after the $\bar{k}$-th, and

$z \notin D_{\mathsf{S}} \setminus T_{\bar{k}}$ yields that $T'_{\bar{k}+1} = T_{\bar{k}} \cup \{z\}$ contains the element $z$ as many times as it appears in $D \cup \{z\}$, i.e., as many times as it appears in $D$ plus one. Therefore $T'_{\bar{k}+1} \not\subseteq D$ and, since the multiset $T'_k$ is increasing with $k$, it must be that $T'_k \not\subseteq D$ for all $k \geq \bar{k} + 1$. This implies $\mathsf{c}_{\mathcal{A}}(D, z) \not\subseteq D$; that is, $\mathsf{c}_{\mathcal{A}}(D, z) \neq V$ for any $V \subseteq D$. $\qquad\square$

We now show that $\mathsf{c}_{\mathcal{A}}$ is non-associative.

**Lemma A.7.** *The compression function $\mathsf{c}_{\mathcal{A}}$ satisfies the non-associativity Property A.2.*

*Proof.* Fix any multiset $D$ of elements of $\mathcal{Z}$ and further let $z_1, \ldots, z_p \in \mathcal{Z}$. We will prove the claim by contrapositive. Towards this goal, assume that $\mathsf{c}_{\mathcal{A}}(D, z_1, \ldots, z_p) \neq \mathsf{c}_{\mathcal{A}}(D)$. We will then show that there exists an $i \in \{1, \ldots, p\}$ such that $\mathsf{c}_{\mathcal{A}}(D, z_i) \neq \mathsf{c}_{\mathcal{A}}(D)$.

We first focus on the executions of $\mathcal{A}(D)$ and $\mathcal{A}(D, z_1, \ldots, z_p)$. Similarly to the proof of Lemma A.6, denote by $T_k, h_k$ and $T_k^{(p)}, h_k^{(p)}$ the multisets $T$ and hypotheses $h$ constructed in $\mathcal{A}(D)$ and $\mathcal{A}(D, z_1, \ldots, z_p)$, respectively, at iteration $k$ ($k = 0$ corresponds to the initialization). Finally, let $D_{\mathsf{S}}^{(p)} = D_{\mathsf{S}} \cup \{z_1, \ldots, z_p\}$. Since $\mathsf{c}_{\mathcal{A}}(D, z_1, \ldots, z_p) \neq \mathsf{c}_{\mathcal{A}}(D)$, also in the present context there must exist a $\bar{k} \geq 0$ such that the executions of $\mathcal{A}(D)$ and $\mathcal{A}(D, z_1, \ldots, z_p)$ match for $k < \bar{k}$ (so that $T_k = T_k^{(p)}$ and $h_k = h_k^{(p)}$ for $k \leq \bar{k}$), but they differ at iteration $\bar{k}$ because, for $k = \bar{k}$, the maximal element computed in $\mathcal{A}(D)$ (line 5 if $\bar{k} \geq 1$ or line 1 if $\bar{k} = 0$) is different from the one computed in $\mathcal{A}(D, z_1, \ldots, z_p)$. That is, $\max_{h_{\bar{k}}}(D_{\mathsf{S}} \setminus T_{\bar{k}}) \neq \max_{h_{\bar{k}}^{(p)}}(D_{\mathsf{S}}^{(p)} \setminus T_{\bar{k}}^{(p)})$. Then, following an argument identical to that in the proof of Lemma A.6, one observes that it must be that $\max_{h_{\bar{k}}^{(p)}}(D_{\mathsf{S}}^{(p)} \setminus T_{\bar{k}}^{(p)}) = z_i$ for some $i \in \{1, \ldots, p\}$ and $z_i \notin D_{\mathsf{S}} \setminus T_{\bar{k}}$.

Now compare the executions of $\mathcal{A}(D)$ and $\mathcal{A}(D, z_i)$, and denote by $T'_k, h'_k$ the multiset $T$ and the hypothesis $h$ constructed in the execution of $\mathcal{A}(D, z_i)$ at iteration $k$ ($k = 0$ still corresponds to the initialization). Further, let $D'_{\mathsf{S}} = D_{\mathsf{S}} \cup \{z_i\}$. One can first observe that also the executions of $\mathcal{A}(D)$ and $\mathcal{A}(D, z_i)$ for $k < \bar{k}$ must be identical. Notice indeed that: i. $T'_0 = T_0 = T_0^{(p)}$ and $h'_0 = h_0 = h_0^{(p)}$; ii. when $T'_k = T_k = T_k^{(p)}$ and $h'_k = h_k = h_k^{(p)}$ for a $k < \bar{k}$, the maximal element selected at iteration $k$ of $\mathcal{A}(D)$ and $\mathcal{A}(D, z_i)$ is the same since this is so for the maximal element of $\mathcal{A}(D)$ and $\mathcal{A}(D, z_1, \ldots, z_p)$ and $D_{\mathsf{S}} \setminus T_k \subseteq D'_{\mathsf{S}} \setminus T'_k \subseteq D_{\mathsf{S}}^{(p)} \setminus T_k^{(p)}$; iii. this in turn gives $T'_{k+1} = T_{k+1} = T_{k+1}^{(p)}$ and $h'_{k+1} = h_{k+1} = h_{k+1}^{(p)}$. At iteration $\bar{k}$, instead, the identity of executions ends, and it must be $\max_{h'_{\bar{k}}}(D'_{\mathsf{S}} \setminus T'_{\bar{k}}) = z_i$ because $z_i$ is maximal over $D_{\mathsf{S}}^{(p)} \setminus T_{\bar{k}}^{(p)}$ and $z_i \in D'_{\mathsf{S}} \setminus T'_{\bar{k}} \subseteq D_{\mathsf{S}}^{(p)} \setminus T_{\bar{k}}^{(p)}$ (surely $z_i$ still belongs to $D'_{\mathsf{S}} \setminus T'_{\bar{k}}$ because $T'_{\bar{k}} = T_{\bar{k}} \subseteq D$ and $D'_{\mathsf{S}} = D \cup \{z_i\} \cup \{\texttt{Stop}\}$). Further, recall that $z_i \notin D_{\mathsf{S}} \setminus T_{\bar{k}}$. Since $\mathcal{A}(D, z_i)$ executes at least one more iteration after the $\bar{k}$-th (because $z_i \neq \texttt{Stop}$) and since the $T'_k$'s are increasing, we have that any $T'_k$ for $k \geq \bar{k} + 1$ is different from any of the $T_k$'s. This implies $\mathsf{c}_{\mathcal{A}}(D, z_i) \neq \mathsf{c}_{\mathcal{A}}(D)$, thus concluding the proof. $\qquad\square$

Given that Properties A.1 and A.2 hold true, and since Property A.3 is assumed, Theorem A.4 applies, yielding $\mathbb{P}\{\underline{\varepsilon}(|\mathsf{c}_{\mathcal{A}}(\boldsymbol{D})|, \delta) \leq \phi_{\mathcal{A}}(\boldsymbol{D}) \leq \bar{\varepsilon}(|\mathsf{c}_{\mathcal{A}}(\boldsymbol{D})|, \delta)\} \geq 1 - \delta$, where $\phi_{\mathcal{A}}(\boldsymbol{D}) = \mathbb{P}\{\mathsf{c}_{\mathcal{A}}(\mathsf{c}_{\mathcal{A}}(\boldsymbol{D}), \boldsymbol{z}) \neq \mathsf{c}_{\mathcal{A}}(\boldsymbol{D}) \mid \boldsymbol{D}\}$. As anticipated, to obtain from this the sought result for the statistical risk, it is enough to show that, given $D$ and the hypothesis $h$ returned by $\mathcal{A}(D)$, every realization $z$ such that $\texttt{Stop} \leq_h z$ also changes the compression, i.e., $\mathsf{c}_{\mathcal{A}}(\mathsf{c}_{\mathcal{A}}(D), z) \neq \mathsf{c}_{\mathcal{A}}(D)$. As a matter of fact, this implies that for every $D$ the event $\{\texttt{Stop} \leq_h z\}$ is a sub-event of $\{\mathsf{c}_{\mathcal{A}}(\mathsf{c}_{\mathcal{A}}(D), z) \neq \mathsf{c}_{\mathcal{A}}(D)\}$, and hence that $R(\boldsymbol{h}) \leq \phi_{\mathcal{A}}(\boldsymbol{D})$ almost surely. Then, Theorem 4.2's claim readily follows since

$$
\begin{aligned}
\mathbb{P}\{R(\boldsymbol{h}) &\leq \bar{\varepsilon}(|\mathsf{c}_{\mathcal{A}}(\boldsymbol{D})|, \delta)\} \\
&\geq \quad \mathbb{P}\{\phi_{\mathcal{A}}(\boldsymbol{D}) \leq \bar{\varepsilon}(|\mathsf{c}_{\mathcal{A}}(\boldsymbol{D})|, \delta)\} \\
&\geq \quad \mathbb{P}\{\underline{\varepsilon}(|\mathsf{c}_{\mathcal{A}}(\boldsymbol{D})|, \delta) \leq \phi_{\mathcal{A}}(\boldsymbol{D}) \leq \bar{\varepsilon}(|\mathsf{c}_{\mathcal{A}}(\boldsymbol{D})|, \delta)\} \\
&\geq \quad 1 - \delta.
\end{aligned}
$$

Thus, we are left to prove the following lemma.

**Lemma A.8.** *Let $D$ be any multiset of elements of $\mathcal{Z}$, $z \in \mathcal{Z}$ be any example, and $(h, T) = \mathcal{A}(D)$. If $z$ is such that $\texttt{Stop} \leq_h z$, then $\mathsf{c}_{\mathcal{A}}(\mathsf{c}_{\mathcal{A}}(D), z) \neq \mathsf{c}_{\mathcal{A}}(D)$.*

*Proof.* We prove the contrapositive and show that, if $z$ is such that $c_{\mathcal{A}}(c_{\mathcal{A}}(D), z) = c_{\mathcal{A}}(D)$, then $z \leq_h$ Stop.

Note that $(h, T) = \mathcal{A}(D)$ means that $T = c_{\mathcal{A}}(D)$ and $h = L([T]_{\mathcal{A}})$ thanks to line 4 in Algorithm 1. Consider the execution of $\mathcal{A}(c_{\mathcal{A}}(D) \cup \{z\})$. By definition of $c_{\mathcal{A}}$, the condition $c_{\mathcal{A}}(c_{\mathcal{A}}(D), z) = c_{\mathcal{A}}(D)$ means that $\mathcal{A}(c_{\mathcal{A}}(D) \cup \{z\})$ terminates with $c_{\mathcal{A}}(D) = T$. Hence, the maximal elements selected by $\mathcal{A}(c_{\mathcal{A}}(D) \cup \{z\})$ are the same as those selected by $\mathcal{A}(D)$. Further, by the properties of max operators, the elements selected by $\mathcal{A}(c_{\mathcal{A}}(D) \cup \{z\})$ and $\mathcal{A}(D)$ match at each iteration. This implies that $\mathcal{A}(c_{\mathcal{A}}(D) \cup \{z\})$ terminates returning $h$, besides $T$. For this to occur, it must be that $\max_h((c_{\mathcal{A}}(D) \cup \{z\} \cup \{\text{Stop}\}) \setminus T) = \text{Stop}$ (line 2). This is equivalent to $\max_h(\{z, \text{Stop}\}) = \text{Stop}$, i.e., $z \leq_h$ Stop. $\qquad\square$

Having proven Lemma A.8, we have completed the derivation of the main result *under the additional assumption that Property A.3 holds true*, an assumption that, however, does not appear in the statement of Theorem 4.2. To remove this assumption, and thus complete the proof of Theorem 4.2, we can proceed as follows.

Augment each random element $z_i$ with a random variable $u_i$ that is uniformly distributed over $[0, 1]$ and independent of $z_i$. Moreover, define the $u_i$'s so that the random elements $\tilde{z}_1, \tilde{z}_2, \ldots, \tilde{z}_N$, where $\tilde{z}_i = (z_i, u_i)$ for all $i$, are independent and identically distributed (i.i.d.). Also, let $\tilde{z} = (z, u)$ be independent of and identically distributed as each $\tilde{z}_i$. Let $\tilde{\mathcal{Z}} = \mathcal{Z} \times [0, 1]$, which is the space where the $\tilde{z}_i$'s take value. For any $\tilde{z} = (z, u) \in \tilde{\mathcal{Z}}$ and $\tilde{z}' = (z', u') \in \tilde{\mathcal{Z}}$ define $\tilde{z} \leq_h \tilde{z}'$ if $z \leq_h z'$ and $u \leq u'$, while define $\tilde{z} \leq_h$ Stop [resp. Stop $\leq_h \tilde{z}$] if $z \leq_h$ Stop [resp. Stop $\leq_h z$]. Also, for any $n$, define $L(\tilde{z}_1, \ldots, \tilde{z}_n) = L(z_1, \ldots, z_n)$, where $\tilde{z}_i = (z_i, u_i)$, $i = 1, \ldots, n$, are elements of $\tilde{\mathcal{Z}}$.

Consider now the executions of $\mathcal{A}$ over multisets of elements of $\tilde{\mathcal{Z}}$ according to the redefinition of $\leq_h$ and $L$, and let $(\tilde{h}, \tilde{T}) = \mathcal{A}(\tilde{D})$, where $\tilde{D} = \{\tilde{z}_1, \ldots, \tilde{z}_N\}$. As is clear, the non-concentrated mass Property A.3 applies for the $\tilde{z}_i$'s and an application of the results obtained so far gives $\mathbb{P}\{\tilde{R}(\tilde{h}) \leq \overline{\varepsilon}(|\tilde{T}|, \delta)\} \geq 1 - \delta$, where $\tilde{R}(h) = \mathbb{P}\{\text{Stop} \leq_h \tilde{z}\}$. On the other hand, by virtue of the redefinition of $\leq_h$ and $L$, one can notice that, for every realization of $\tilde{D}$, the execution of $\mathcal{A}(\tilde{D})$ matches step by step that of $\mathcal{A}(D)$, with the sole difference that elements in $T$ that are otherwise indistinguishable are incorporated in $\tilde{T}$ with their "$u$" counterparts (and the highest values are selected first). Thus, the multiset $T$ is equal to the multiset of the "$z$" components of the elements of $\tilde{T}$, yielding $|\tilde{T}| = |T|$. Moreover, since the addition of elements in $T$ and $\tilde{T}$ occurs in the same order, we also have that $\tilde{h} = h$. Finally, noticing that $\tilde{R}(h) = R(h)$ because $\tilde{z} \leq_h$ Stop $\iff z \leq_h$ Stop, one has that

$$\mathbb{P}\{R(h) \leq \overline{\varepsilon}(|T|, \delta)\} = \mathbb{P}\{\tilde{R}(\tilde{h}) \leq \overline{\varepsilon}(|\tilde{T}|, \delta)\} \geq 1 - \delta,$$

and this concludes the proof of Theorem 4.2.

*Remark* A.9 (On the tightness of Theorem 4.2). *Although both upper and lower bounds that hold with confidence $1 - \delta$ are available for the probability of change of compression, only the upper bound ports over to the statistical risk, because any realization for which Stop $\leq_h z$ (i.e., a realization that is inappropriate) is also a realization that changes the compression, but the opposite does not hold. Given that the lower and the upper bounds are provably close to each other even for relatively small values of $N$, this means that $\overline{\varepsilon}(|T|, \delta)$ is an accurate evaluation of the probability of change of compression, but it can be loose for the statistical risk whenever this deviates from the probability of change of compression. Interestingly, this is the only source of looseness in the evaluation of the risk provided by Theorem 4.2.*

### A.3 An extension of P2L: adding multiple examples in $T$ simultaneously

A simple extension of the meta-algorithm P2L described in Algorithm 1 consists in adding at every iteration multiple, say $R > 1$, examples in $T$ at a time. A possible implementation is then formalized in Algorithm 2 below, where $\max_h^R(U)$ returns the $R$ maximal points (the maximum, the 2nd maximum, ..., the $R$-th maximum) of $U$ according to the total order $\leq_h$ (if $U$ has less than $R$ elements, then the whole $U$ is returned).

---
**Algorithm 2** $\mathcal{A}^R(D)$ – Meta-algorithm P2L with $R$ examples added at a time
---
1: **Initialize:** $T = \emptyset$, $h = h_0$, $\bar{z}_1, \ldots, \bar{z}_R \leftarrow \max_h^R(D_{\mathtt{S}})$
2: **while** $\bar{z}_i \neq \mathtt{Stop}$ for all $i = 1, \ldots, R$ **do**
3:     $T \leftarrow T \cup \{\bar{z}_1, \ldots, \bar{z}_R\}$            $\triangleright$ Augment $T$
4:     $h \leftarrow L([T]_{\mathcal{A}})$            $\triangleright$ Learn hypothesis
5:     $\bar{z}_1, \ldots, \bar{z}_R \leftarrow \max_h^R(D_{\mathtt{S}} \setminus T)$            $\triangleright$ Compute max
6: **end while**
7: $T \leftarrow T \cup \{\bar{z}_i : \bar{z}_i \geq_h \mathtt{Stop}\}$            $\triangleright$ Complete $T$
8: **return** $h, T$            $\triangleright$ Hypothesis $h$ and multiset $T$
---

Interestingly, Theorem 4.2 continues to hold true (with no modification in the statement) also when the pair $h, T$ is the output of Algorithm 2 instead of Algorithm 1. To prove this, it is enough to observe that Algorithm 2 is completely equivalent to the following sequential Algorithm 3, which is identical to Algorithm 1, so that only one point is added to $T$ at each iteration, except for the fact that $h$ is updated every $R$ iterations ($\mathrm{mod}(\mathrm{iter}, R)$ is the remainder of the division of iter by $R$). This equivalence secures the result because the proof of Theorem 4.2 can be repeated for Algorithm 3 word for word, without any modification.

---
**Algorithm 3**
---
1: **Initialize:** $T = \emptyset$, $h = h_0$, $\bar{z} = \max_{h_0}(D_{\mathtt{S}})$, iter $= 0$
2: **while** $\bar{z} \neq \mathtt{Stop}$ **do**
3:     iter $\leftarrow$ iter $+ 1$
4:     $T \leftarrow T \cup \{\bar{z}\}$            $\triangleright$ Augment $T$
5:     **if** $\mathrm{mod}(\mathrm{iter}, R) = 0$ **then**
6:         $h \leftarrow L([T]_{\mathcal{A}})$            $\triangleright$ Learn hypothesis
7:     **end if**
8:     $\bar{z} \leftarrow \max_h(D_{\mathtt{S}} \setminus T)$            $\triangleright$ Compute max
9: **end while**
10: **return** $h, T$            $\triangleright$ Hypothesis $h$ and multiset $T$
---

As is clear, Algorithm 1 is a particular case of Algorithm 2 since the former is re-obtained from the latter when $R = 1$. We notice that adding more than one example to $T$ at every iteration might or might not have a beneficial effect on P2L. Indeed, on one hand, adding more examples simultaneously might result in building a better $h$ earlier on (and save computational time). However, by introducing examples in groups (and thus not allowing for the fine-grained choice of adding them one-by-one) might also result in a larger $T$, which could worsen the final bound on the risk. In this regard, a modulation of $R$ across iterations seems to be a promising line of investigation, which however is beyond the scope of the present paper and is therefore left for future work.

### A.4  A discussion on the P2L framework in retrospect

A close inspection of the proofs of the key Lemmas A.6 and A.7 reveals that these two lemmas strongly rely on the fact that the $\max_h$ operator is a compression function from many elements to just one with the following properties: i) it is preferent (as a matter of fact, for any multiset $A$ and additional element $b$, either $\max_h(A \cup \{b\}) = b$ or $\max_h(A \cup \{b\}) = \max_h(A)$); and ii) it is non-associative ($\max_h(A \cup \{b_i\}) = \max_h(A)$ for all $i \in \{1, \ldots, p\}$ implies that there is an element $\bar{a}$ of $A$ such that $c \leq_h \bar{a}$ for all $c \in A \cup \{b_1, \ldots, b_p\}$ and, therefore, $\max_h(A \cup \{b_1, \ldots, b_p\}) = \max_h(A)$).

Importantly, these are the only two properties of $\max_h$ that are used and, therefore, one could have formulated Algorithm 1 in an (apparently, see below) more general form: instead of introducing a hypothesis-dependent total ordering $\leq_h$, one might have considered in its place a hypothesis-dependent preferent and non-associative compression function from many to one example. *Mutatis mutandis*, Theorem 4.2 and the arguments to prove it would have remained the same. This generalization extends to the context in which multiple examples are selected at a time as suggested in Appendix A.3.

We feel advisable to also notice that the above generalization is indeed a real generalization only in the case of selection of multiple example at a time. Instead, in the context of one example selected at a time the alternative setup illustrated above turns out *not* to be more general than that of Section 4 (this is why we wrote "apparently" in parenthesis). In fact a converse property holds: any compression function $\mathtt{w}$ from many examples to one that is preferent and non-associative always defines a total ordering whose notion of $\max$ coincides with the compression function itself. This fact follows from the following proposition (see a few lines down for a proof).

**Proposition A.10.** *Any preferent compression function $\mathtt{w}$ from many examples to one defines a total ordering $\leq_{\mathtt{w}}$ for which it holds that $\mathtt{w}(a_1, \ldots, a_n) = \max_{\mathtt{w}}(a_1, \ldots, a_n)$.*

Note that, as a byproduct of Proposition A.10, we have that any preferent compression function $\mathtt{w}$ from many examples to one is also non-associative (since $\max_{\mathtt{w}}$ is non-associative). Therefore, the notion of $\max$ from a total ordering and that of preferent and non-associative compression function from many examples to one are equivalent. Interestingly, this equivalence crucially relies on the fact that the compression selects only one example, while, provably, the class of preferent and non-associative compression functions offers more freedom than using a total ordering when selecting more than one example at a time. This is important for the development of alternative implementations of Algorithm 2, the version of P2L where multiple examples are selected at a time.

*Proof of Proposition A.10.* For any $a$ and $b$ define $a \leq_{\mathtt{w}} b$ if $\mathtt{w}(a, b) = b$ (and thus $b \leq_{\mathtt{w}} a$ if $\mathtt{w}(a, b) = a$). To show that $\leq_{\mathtt{w}}$ is indeed an ordering, we have to prove that $\leq_{\mathtt{w}}$ enjoys the *reflexive*, *antisymmetric*, and *transitive* properties.

   a. Since $\mathtt{w}$ always selects one element, we have that $\mathtt{w}(a, a) = a$, i.e., $a \leq_{\mathtt{w}} a$, which is the *reflexive* property.

   b. If $a \leq_{\mathtt{w}} b$ and $b \leq_{\mathtt{w}} a$, then $\mathtt{w}(a, b) = b$ and $\mathtt{w}(a, b) = a$. Thus, $a \leq_{\mathtt{w}} b$ and $b \leq_{\mathtt{w}} a \implies a = b$, which is the *antisymmetric* property.

   c. Suppose that $a \leq_{\mathtt{w}} b$ and $b \leq_{\mathtt{w}} c$, which corresponds to $\mathtt{w}(a, b) = b$ and $\mathtt{w}(b, c) = c$. Since $\mathtt{w}$ is preferent, it must be that $\mathtt{w}(a, b, c) = c$ (indeed, $\mathtt{w}(a, b, c)$ cannot be $a$ because $a$ is not selected by $\mathtt{w}(a, b)$; it cannot be $b$ either because $b$ is not selected by $\mathtt{w}(b, c)$). Then, an application of preference again gives $\mathtt{w}(a, c) = c$. This means that $a \leq_{\mathtt{w}} b$ and $b \leq_{\mathtt{w}} c$ implies that $a \leq_{\mathtt{w}} c$, which is the *transitive* property.

We next prove that the $\max_{\mathtt{w}}$ operator associated to $\leq_{\mathtt{w}}$ corresponds indeed to $\mathtt{w}$; that is, it holds that $a_i \leq_{\mathtt{w}} \mathtt{w}(a_1, \ldots, a_n)$ for all $i \in \{1, \ldots, n\}$. The proof is by contrapositive. Suppose thus that $\mathtt{w}(a_1, \ldots, a_n) = a_i$ for some $i$, but $a_i <_{\mathtt{w}} a_j$ (i.e., $a_i \leq_{\mathtt{w}} a_j$ and $a_i \neq a_j$) for some $j$. Condition $a_i <_{\mathtt{w}} a_j$ means that $\mathtt{w}(a_i, a_j) = a_j$. By preference, adding another element $a_k$ to $a_i$ and $a_j$ would give $\mathtt{w}(a_i, a_j, a_k) \neq a_i$. But then, preference again would give that the compression of $a_i, a_j$, and $a_k$ plus a fourth element cannot be $a_i$, and so forth and so on. Thus, proceeding iteratively would eventually lead to $\mathtt{w}(a_1, \ldots, a_n) \neq a_i$, which is a contradiction. $\qquad\square$

# B   Additional material for the application to MNIST Classification

## B.1   Implementation details for SGD and PAC-Bayes

**GD & test-set approach.** Here we follow a classical test-set approach, whereby a fraction of the dataset is used for training, and the remaining portion is sacrificed to certify the quality of the resulting model. We experiment with different sizes of these fractions. As for training, we use GD with momentum and fix the number of training epochs to 200. To compute the generalization bound, we use the tightest-known Binomial test-set bound[10] of (Langford, 2005)[Thm 3.3] with a number of samples equal to $N$ minus those used for training. That is, letting $\tilde{N}$ be the number of examples out

---

[10]The Binomial test set bound is tighter then the commonly utilized Chernoff bound.

of $N$ used for training and $\tilde{h} = GD(z_1, \ldots, z_{\tilde{N}})$ be the returned hypothesis, we have that

$$R(\tilde{h}) \quad \leq \quad \max \left\{ p \in [0,1] : \sum_{j=0}^{k} \binom{N-\tilde{N}}{j} p^j (1-p)^{N-\tilde{N}-j} \geq \delta, \right. \tag{4}$$

$$\left. \text{where } k \text{ is the number of } z_j, j = \tilde{N}+1, \ldots, N, \text{ misclassified by } \tilde{h} \right\}$$

with high confidence $1 - \delta$. We also perform a grid search over the following set of parameters: learning rate in $[0.001, 0.005, 0.01]$, momentum in $[0.9, 0.95]$, dropout probability in $[0.01, 0.05, 0.1, 0.2]$. We then select the combination of parameters that return the lowest generalization bound.[11]

**PAC-Bayes approach.** Here we learn a model and provide generalization bounds based on the recent results presented in (Clerico et al., 2022; Perez-Ortiz et al., 2021), which build on earlier work in (Dziugaite & Roy, 2017) while providing tighter generalization bounds. Specifically, we utilize the same PAC-Bayes implementation of (Perez-Ortiz et al., 2021) adapted to the considered binary classification problem, with one main difference: instead of training by optimizing a surrogate of the generalization bound as in (Perez-Ortiz et al., 2021), we optimize the exact bound as proposed in (Clerico et al., 2022, Eq 4.a), which provides the sharpest results to date.[12] As the PAC-Bayes approach requires a prior distribution over the networ weights to start, we split, as it is customary, the training dataset in two and use the first part to pretrain the prior through SGD, while the remaining data are used to obtain the posterior distribution (contrary to P2L and GD+test-set where the final hypothesis consists of a deterministic network, the PAC-Bayes approach is forced to work with stochastic networks). We experiment with different sizes of the data portion to be used to pretrain the prior. In running PAC-Bayes we employ a batch size of 250, 100 prior and posterior training epochs, and $10^{-6}$ as lower bound for the minimum probability of the softmax output. We also perform a grid search over the remaining parameters: prior standard deviation in $[0.01, 0.02, 0.03, 0.04, 0.05, 0.1]$; dropout probability in $[0.01, 0.05, 0.1, 0.2]$; prior/posterior learning rate in $[0.001, 0.005, 0.01]$; prior/posterior momentum in $[0.9, 0.95]$. The final bound we present is that in (Perez-Ortiz et al., 2021)[Sec 6.2], by using 10000 Monte Carlo samples from the learned distribution over the models to approximate the average empirical risk. When we experiment with $\delta = 0.035$ ($\delta = 0.001$), we use a confidence of 0.025 (0.00072) for the KL bound and of 0.01 (0.00028) for Monte Carlo sampling as done in (Perez-Ortiz et al., 2021)[Sec 6.2]. We apply here the same philosophy as described in Footnote 11 regarding not introducing a union bound correction. The post-training performance is evaluated again by sampling the distribution over the network weights, computing the empirical misclassification on the full test dataset, and averaging over 10000 samples of the weights. As a matter of fact, the generalization bounds derived through the PAC-Bayes approach apply precisely to this quantity.

## C   Some examples when the inner algorithm L has already in itself a compression scheme

In this section, also prompted by a reviewer, we perform some synthetic numerical experiments to investigate how P2L performs when the inner algorithm $L$ already exhibits a compression in itself; specifically, we consider SVM (Support Vector Machine, (Cortes & Vapnik, 1995)) and SVR (Support Vector Regression, (Smola & Schölkopf, 2004)) as inner algorithms for P2L and compare the returned output with that output by SVM and SVR alone. The so-called support vectors returned by SVM and SVR are known to compress the original dataset in a way that is informative for the underlying learning problem. With these experiments, we want to verify whether P2L returns the same or a similar compression.

---

[11]Formally, we should introduce a union bound correction because we perform a grid search over multiple hyper-parameters. Notably, this correction would enlarge (i.e., worsen) the risk bound. However, as the union bound might be conservative, we do not pursue this approach and give this advantage to both the test-set and, later, the PAC-Bayes approach.

[12]The research in the field of PAC-Bayes is extremely active, and we would like to acknowledge paper (Wu & Seldin, 2022), which has introduced improved PAC-Bayes bounds based on novel concentration inequalities for random variables with ternary values. However, in a binary setup (as in the present application), (Wu & Seldin, 2022) itself notices that there is no advantage over the existing literature. Thus, to the best of our knowledge, the bounds proposed by (Clerico et al., 2022) are currently the tightest for our problem.

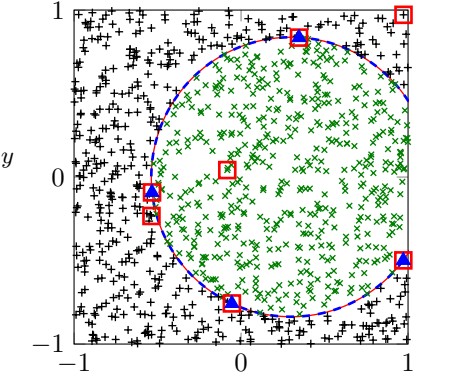 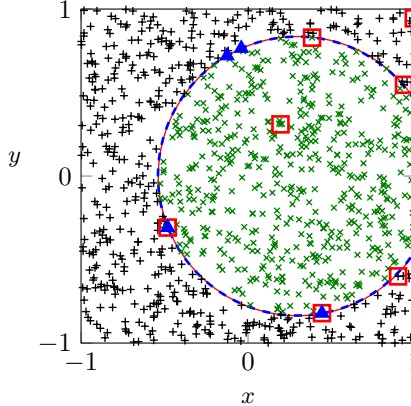

Figure 5: Comparison of learned classifiers and compressed sets returned by SVM and P2L+SVM. Markers $+$, $\times$ represent examples with labels $\pm 1$. The dashed blue and solid red lines (**- -**, **—**), which are completely overlapped, represent the classifier returned by SVM and P2L+SVM. The blue triangles and red squares ($\blacktriangle$, $\square$) represent the support vectors returned by SVM and the compressed set $T$ returned by P2L+SVM.

In a classification problem, we consider datasets with $N = 1000$ examples $(x, y)$, where instances $x$ are uniformly sampled from $[-1, 1]^2 \subseteq \mathbb{R}^2$ and labels are 1 if $(x_1 - 0.3)^2 + x_2^2 \leq 0.7$ and $-1$ otherwise. SVM (either when run over the whole dataset or when called by P2L as inner algorithm) is implemented with kernel $k(x, \tilde{x}) = x \cdot \tilde{x}' + \|x\|^2 \cdot \|\tilde{x}\|^2$, for which the two classes defined above are separable. P2L is run by selecting as $h_0$ the classifier corresponding to setting all the parameters in the SVM parameterization to 1. As for $\leq_h$, we consider the order induced by the distance in the lifted feature space of examples from the separating hyperplane corresponding to $h$, multiplied by 1 when examples are misclassified and by $-1$ otherwise (this yields that the worst examples are those misclassified and furthest away from the separating hyperplane). Distance equal to 0 is taken as the threshold for appropriateness, which corresponds to consider a classifier $h$ appropriate for an example $(x, y)$ when $(x, y)$ is not misclassified by $h$.

Figure 5 displays the results for two realizations of the considered classification problem, which are representative of what we have observed in multiple trials. Each plot depicts: the examples in the instance domain as markers $+$ (label 1), $\times$ (label $-1$); the separation boundaries of the SVM and of the P2L+SVM classifiers (dashed blue and solid red lines, which are completely overlapped); the support vectors returned by SVM (red squares) and the examples in $T$ returned by P2L+SVM (blue triangles).

In the left panel of Figure 5, the compressed multiset $T$ returned by P2L+SVM contains all the support vectors of SVM with the addition of few other examples, which are incorporated in $T$ during the first iterations, while P2L is still exploring the problem. As it is clear, by the definition of support vectors, the classifier returned by P2L+SVM coincides with the classifier obtained by running SVM. In the right panel of Figure 5, only part of the support vectors are also elements of $T$. Nonetheless the classifiers obtained with P2L+SVM and SVM are still the same (at least up to a negligible approximation). Hence, it appears that P2L+SVM compensates the deficiency of support vectors with other alternative examples that are anyway capable to recover the SVM classifier. The amount of support vectors appearing in $T$ depends on the realization of the dataset, but in all our trials the classifier returned by P2L+SVM matches that obtained by running SVM on the same dataset.

As for SVR, we consider the same regression problem as in Section 6 when $N = 200$. SVR is run using a radial basis function $k(x, \tilde{X}) = \exp(-0.1\|x - \tilde{x}\|^2)$ as kernel, tube size equal to $0.1$, and coefficient $C = 1$ ($C$ modulates the trade-off between the flatness of the predictor and the penalty for deviations from the predictor larger than $0.1$). P2L instead starts with a predictor $h_0$ corresponding to setting all the parameters in the SVR parameterization to 0, while we consider the same total order $\leq_h$ of Example 3.2 with $\gamma = 0.1$. In this context, $h$ is appropriate for an example $(x, y)$ if the deviation of the output $y$ from the prediction corresponding to $x$ is no more than $0.1$.

Figure 6 displays the results obtained for one dataset, which is also representative of other realizations of data. In the plot, examples are represented as black crosses, while the dashed blue and red lines,

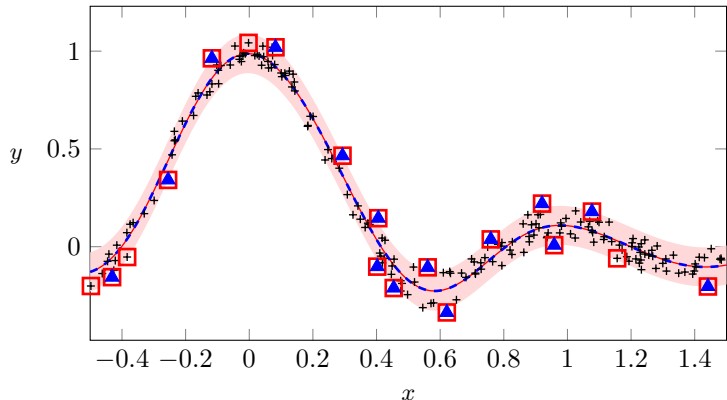

Figure 6: Comparison of learned predictors and compressed sets returned by SVR and P2L+SVR. Markers + represent examples. The dashed blue and solid red lines (--, −), which are completely overlapped, represent the predictor returned by SVR and P2L+SVR. The red shaded region depicts a tube of radius 0.1 around such predictors. The blue triangles and red squares (▲, □) represent the support vectors returned by SVR and the compressed set $T$ returned by P2L+SVR.

which are completely overlapped, depict the predictors returned by SVR and P2L+SVR respectively. The support vectors returned by SVR are displayed by blue triangles, while the examples in $T$ returned by P2L+SVR by red squares.

Similarly to the SVM case, in all our trials the predictor returned by P2L+SVR is identical to the predictor obtained by running SVR on the same dataset. In the present example, however, it is always the case that the compressed multiset $T$ returned by P2L+SVR contains all the support vectors of SVR, with the addition of few other examples incorporated in $T$ during the first iterations during which P2L is still exploring the problem.

