# OpenReview forum: "The Pick-to-Learn Algorithm: Empowering Compression for Tight Generalization Bounds and Improved Post-training Performance"
_NeurIPS.cc/2023/Conference — NeurIPS 2023 spotlight_

### Official Review · Reviewer_q61g · 2023-06-30

**Soundness:** 4 excellent
**Presentation:** 3 good
**Contribution:** 3 good
**Rating:** 6
**Confidence:** 4

**Summary:**

This paper introduces a meta-learning algorithm which compresses a training set $D$ of size $|D|$ into a smaller training set $T\subset D$ of size $|T|$. The algorithm can be defined for any *base learner*, i.e. any training strategy which outputs a trained hypothesis $h$ when given a training set. The set $T$ is selected in such a way that the performance of the base algorithm trained on $T$ and evaluated on $D$ is better than a predetermined threshold. Relying on some powerful recent results from [1], the authors show that one can obtain tight generalization bounds for the result of the algorithm expressed as a function of the size  ratio $|T|/|D|$: intuitively, if the model is able to compress the dataset $D$ into a much smaller dataset $|T|$ whilst maintaining the performance of the base learner trained on $T$, it must be that the sample size is already sufficient to train the model well. Thus, the results can be interpreted as a middle way between on the one hand  the traditional approach to generalization bounds via direct study of the function class searched by a given algorithm, and on the other hand,  the even simpler *test-set bounds*[2]. The function of the ratio $|T|/|D|$ which gives a bound on the probability of error is a complex implicitly defined function inherited from [1] with a very mild dependence on the failure probability $\delta$.

The algorithm functions as follows: start with an initial hypothesis $h_0$, then find the element of the training set $D$ which is the least well explained by $h$ (for instance, the one with the largest loss function value), and include this element in the set $T$. Retrain the hypothesis based on $T$ and repeat the procedure until either every element of the set $D\setminus T$ has a loss function value below a given predetermined threshold.

The proof of the generalization bounds is a reasonably direct consequence of Theorem A.4., which is a powerful result from [1] concerning compression schemes. The key is to show that the compression function defined by the algorithm (the function which compresses $D$ into $T$ satisfies the properties of *preference* and *non associativity*, which are proved in Lemmas A.5. and A.6 respectively, and that the failure event which is bounded in Theorem A.4, which is the *probability of change of compression* is implied by the failure of prediction event (this is shown in Lemma A.7.). A technical condition of non concentrated mass which is required to be satisfied for Theorem A.4 to hold, is also removed by a straightforward argument augmenting each datapoint by a continuous random variable which doesn't affect the loss function/order. It is also shown (Proposition A.9) that any compression scheme which outputs a single element and is preferent must correspond to a maximum function with respect to some well defined notion of order.

Experiments on the MNIST dataset and on a synthetic regression dataset demonstrate that the proposed algorithm yields superior generalization performance and tighter bounds than the Pac-Bayesian baseline, and proide slightly inferior bounds compared to a test-set approach. It is still worth noting nonetheless that the test performance is still superior to the test set approach due to a better indirect exploitation of the whole training set in the compression step.


==============Post-rebuttal==========

As seen in the comments below, my discussion with the authors and the extra material promised (especially the experiments in the rebuttal concerning support vectors) have increased my opinion of the paper, resulting in me raising my score to 6. I believe this paper is above the threshold and tackles very interesting questions, with the main downside being a relatively small amount of material and original proofs (which should not necessarily disqualify a paper for publication).

=====


**References**

[1] Marco C. Campi, Simone Garatti. Compression, Generalization and Learning. ArXiv 2023.
[2] Langford, J. and Shapire, R. Tutorial on practical prediction theory for classification. JMLR 2017.

**Strengths:**

1. This is a very interesting direction, not just in terms of providing generalization bounds (which is the main aspect presented in the paper) but also in terms of potential practical interpretation: this is a simple algorithm that, in principle, allows one to compress a dataset into a much smaller one, allowing us to derive interpretable information about which datapoints are key to the training procedure.

2. This is a novel application of a mathematical result from [1] into a machine learning context, and has a distinctly novel flavor compared to many existing generalization bounds.

3. The paper is relatively well written and a pleasure to read. The proofs are clean despite the fact that they must have been a bit annoying to write: It is quite trivial to convince oneself that the results (lemmas A.5, A.6, A.7 and Proposition A.9.) hold, but not so much fun to write down explicitly.

**Weaknesses:**

1. The results are essentially straightforward applications of existing results from [1], there isn't really any non trivial difficulty that had to be vanquished in the proofs. In short, this paper isn't a lot of work from the authors.

2. The experimental evaluation is still relatively preliminary, there is so much more to explore: Other datasets and architectures, better baselines, and more importantly, the implications in terms of interpretability. For instance, the following follow-up experiments could be performed:

2.1 Compare to more baselines on more datasets (Cifar, other Pac Bayesian bounds). In footnote 7 on page 7, the authors claim that "Pac-Bayesian approaches have been developed only for linear regression problems". This is a highly doubtful or (at best misleading) statement. In addition, generalization bounds based on function class capacity could also be evaluated in the synthetic regression dataset.

2.2 (most important) Investigate the interpretability of the results: for instance, it would be interesting and rather key to evaluate the method on a synthetic binary classification dataset with a simple kernel method. It seems that in an ideal situation, the set $T$ should eventually correspond to, or at least strongly overlap, the set of support vectors. Even in the case of MNIST, it would be very interesting to visualize the chosen datapoints and see if they have something qualitatively different from the non-chosen ones.

3 A very big issue is that the algorithm, as presented, is not that applicable in practice. Indeed, to make the algorithm work in the examples considered, the authors needed to pretrain the model on a significant proportion of the training set. In particular, the algorithm cannot select which datapoints in the set used for pretraining are more important than others. Since the algorithms used rely on gradient descent and the authors interpret the call to the base learner as a single gradient step, the initialization is key. It remains to be seen whether the algorithm in its pure form, with a base learner that performs exact empirical risk minimization on $T$, can work in any practical scenario. At the very minimum, it should be checked whether training the network on $T$ from random initialization (rather than pretraining it to obtain $h_0$ and then continuing training with $T$, which is what is done here) yields comparable performance.

3.2 (related to 3) It seems like the extension to the case where several datapoints are introduced into $T$ simultaneously (perhaps in a hierarchical way) would not be too much to ask in a first submission, since it is key to solving the problem of the overreliance on initialization.

4. I feel like the description of the previous results could be more extensive, for the benefit of the reader. For instance, the following things could improve the paper's reach and interest to a broader audience:

4.1 Explain the gist of the proof of theorem A.4., at least in terms of intuitive reason why the function $\Psi$ from line 187 appears.

4.2.  Write down the result (theorem 3.3) from [2] which is used to evaluate the test set bounds and discuss it.

4.3 In line 472, the sentence "See also Section 4 in [1]" appears to suggest that the connection between the probability of change of compression and the classification/regression error is already established in [1]. If that is the case, how much of the present paper is truly not covered or implied by the discussion in [1]?

5. I feel like a more detailed description of existing literature on training set compression schemes and how they relate to the present method is needed. It is hard to believe that no such literature exists.




**Minor comments/typos**

The concept of "probability of change of compression" is frequently used in the main paper (cf., e.g., line 341, but it is only defined in the appendix (455-457)

Line 28: "the precision of available bounds is much problem dependent" ===>  "the precision of available bounds is highlyproblem dependent"

line 64: "is laying the groundwork" ==> "lays the groundwork"

line 110 "is add to"  ===> "is added to"

Lines 114 and 115: "enough appropriate" ==> "appropriate enough"

Line 175:  "an hypothesis" ==> "a hypothesis"

Line 280: "fits well the data" ==> "fits the data well"

lines 480 and 560: I would use "remove the condition" instead of "release the condition"









**References**

[1] Marco C. Campi, Simone Garatti. Compression, Generalization and Learning. ArXiv 2023.
[2] Langford, J. and Shapire, R. Tutorial on practical prediction theory for classification. JMLR 2017.

**Questions:**

Do you think your algorithm will be able to recover support vectors in a simple linear classification problem?

In Table 1 on page 7, which proportion of the training set is used for pretraining?

**Limitations:**

Mostly the reliance on intialization, the limited evaluation of interpretability and the coverage only of the case where a single element is added to the set $T$ at each iteration. See  "weaknesses" for more details".

Assuming there is really no comparable result in the literature (i.e. no generalization bounds expressed in terms of the success of a compression method), this is still a **very interesting paper opening up a new direction**. However, the amount of content included in this first contribution is surgically small.

---

> ### Author Rebuttal · Authors · 2023-08-09
>
> Many thanks for the positive review and constructive comments. In addressing them, we have produced additional material (see attached pdf and below) that we believe will improve the quality of the manuscript.
>
> **Weaknesses**
>
> * 2.1. We concur that our sentence on PAC-Bayes methods in regression problems can be misleading and aim to remove it. We will also evaluate generalization bounds based on class capacity as suggested.
>
> * 2.2. Many thanks for this valuable comment that touches upon an important aspect. We are already able to offer some insights. Specifically, we ran: i) SVM and P2L+SVM as inner algorithm on a (new) synthetic 3d linearly separable classification problem; ii) SVR using RBF kernels and P2L+SVR as inner algorithm on the regression problem presented in Section 6. Interestingly, in both cases, the behaviour is that expected by the Reviewer: the compressed dataset T eventually contains all the support vectors with the addition of few other examples that are added to T in the first iterations, while the algorithm is still learning what are the significant examples. Please see Figures 1 and 2 in the attached PDF. We will add this material to the paper along with some considerations on the chosen examples in the MNIST classification problem.
>
> * 3.&nbsp;We would like to clarify two important points.
>   * First, the inner algorithm, referred by the reviewer as “base learner” (denoted by $L$ in the paper), does not perform a single step of gradient descent, but instead iterates gradient descent until convergence, i.e., “performs empirical risk minimization on T”. As such, the algorithm is given enough freedom to move towards a more informative hypothesis, as guided by the points in T. This is why in the numerical examples the post-training performance of P2L is independent of the portion of data used for pre-training (Fig 2 in manuscript, red line). We apologies if this was not clear -- we will do so in the final version.
>   * Second, we observe that P2L works also without the use of pretraining, see for example the application to synthetic regression in Figure 4 (top panels, red curve at abscissa equal to zero). The motivation for introducing the pre-training is solely that of improving the resulting bound to the risk. Indeed, when starting from an educated guess (i.e., using some data to pretrain $h_0$), the choice of “worst misclassified” point is more meaningful allowing the algorithm to terminate with a smaller set T and thus providing better generalization bounds. Still, good results can be obtained without pre-training and we plan to include evidence for this also for the MNIST example.\
> In this context, it is also worth remarking that the choice of pre-training factor can be thought of as a hyper-parameter, for which our generalization bound provide a methodology to make an optimal choice. Such optimal choice is application specific (e.g., 50% for MNIST, while only 10% for the regression problem).
>
> * 3.2. We thank the Reviewer for their interesting comment. As for the possibility of introducing datapoints in T simultaneously, we note that this can be readily achieved and that our methodology directly accommodates this extension.\
> While we plan to include a discussion on this point in the final version of the manuscript, we observe that adding more than one point to T might or might not have a beneficial effect on P2L. Indeed, on one hand, adding more datapoints might allow to build a better estimate of $h_0$ earlier on. However, by introducing these datapoints in groups (and thus not allowing for the fine-grained choice of adding them one-by-one) might also result in a larger set T, which could worsen the final bound.\
> To illustrate this, we have run the same synthetic regression problem considered in Section 6 (with no pretraining) comparing the case in which, at each iteration, we select only the one/two/three/four/five worst datapoints. The results are presented in Figure 3 in the attached PDF and showcase how, for the problem considered, adding a single data-point at a time is optimal.
>
> * 4.1 and 4.2. We will do so.
>
> * 4.3. In [1] it is only observed that whenever inappropriateness implies a change of compression, then the probability of inappropriateness (the risk) is no bigger than the probability of change of compression. To use this result, however, one has then to show that inappropriateness indeed implies a change of compression, a property which is not straightforward since it is not satisfied by many learning algorithms. The fact that this property always applies for the P2L algorithm is proved here for the first time (Lemma A.7).
> In general, apart from Section A.1, which is a summary of results used in this paper, all the material is new.
>
> * 5.&nbsp;We will improve the description of the existing literature. In particular, we will mention the literature referred to as “data compression” that aims at reducing a given dataset for computational purposes [Toneva et al]. However, these works differ from our since they are not amenable of the generalization bound. We will also contrast our work with existing compression schemes, which include [1]. These works provide generalization bounds but are typically applied to learning algorithms that have built-in compression properties, e.g., SVM. Our contribution departs significantly from them by offering a methodology to enforce a (preferent) compression also when starting from algorithms that do not have this property, e.g., neural nets. To the best of our knowledge this is the first contribution in this space.
>
> * Minor comments/typos: Many thanks for your accurate reading of our paper. All the suggested modifications will be implemented.
>
> **Questions**
>
> * “Do you think …”: Please, see the response to your point 2.2
>
> * “In Table 1 …”: The Table reports the results for pre-training proportions leading to the lowest risk bounds for each method, i.e., 50% for P2L, 70% for SGD+test-set, 60% for PAC-Bayes.

---

> > ### Comment · Reviewer_q61g · 2023-08-13
> > **Thanks for the rebuttal; proof for introducing many datapoints at a time?**
> >
> > Thanks for the very detailed rebuttal. I especially liked the graphs in the PDF showing the interpretability of the chosen points, and I think including this in the final version will indeed substantially improve the quality of the manuscript.
> >
> > Many thanks also for the clarification regarding the number of gradient steps and the promise to incorporate this in the final version.
> >
> > Regarding the addition of several points at the same time (my comment 3.2), I acknowledge your answer and I agree that doing so may actually worsen performance. It is a good idea to incorporate your comments in the final version. However, you also claim that the same techniques can be applied to prove an analogous result when you incorporate several points at the same time. How easy is it? Can you add the complete proof in a pdf and then in the appendix of the final paper?
> >
> > Regarding 4.1, you promised to explain the gist of the proof in the final version. Could you do so here as well?

---

> > > ### Author Response · Authors · 2023-08-18
> > > **Proof + gist of Theorem A.4**
> > >
> > > * Regarding the addition of several points: Many thanks for asking. Interestingly, a simple modification of the original algorithm P2L is sufficient to obtain the desired result, without having to modify any of the proofs. In the following, we describe the modified algorithm which allows one to incorporate $R$ points at every iteration:\
> > > &nbsp;&nbsp;&nbsp;&nbsp;&nbsp;&nbsp; 1. Initialize $T =\emptyset$, $h = h_0$, $z_1,\dots,z_R = \max^R_{h} (D_s)$\
> > > &nbsp;&nbsp;&nbsp;&nbsp;&nbsp;&nbsp; 2. *while* $z_i \neq$ Stop for all i, do\
> > > &nbsp;&nbsp;&nbsp;&nbsp;&nbsp;&nbsp; 3. &nbsp;&nbsp;&nbsp;&nbsp;&nbsp;&nbsp; $T\gets T\cup${$z_1,\dots,z_R$}\
> > > &nbsp;&nbsp;&nbsp;&nbsp;&nbsp;&nbsp; 4. &nbsp;&nbsp;&nbsp;&nbsp;&nbsp;&nbsp; $h\gets L([T]_A)$\
> > > &nbsp;&nbsp;&nbsp;&nbsp;&nbsp;&nbsp; 5. &nbsp;&nbsp;&nbsp;&nbsp;&nbsp;&nbsp; $z_1,\dots,z_R \gets \max^R_h (D_s)$\
> > > &nbsp;&nbsp;&nbsp;&nbsp;&nbsp;&nbsp; 6. *end while*\
> > > &nbsp;&nbsp;&nbsp;&nbsp;&nbsp;&nbsp; 7. $T\gets T\cup${$z_i : z_i\ge$ Stop}\
> > > &nbsp;&nbsp;&nbsp;&nbsp;&nbsp;&nbsp; 8. Return $h,T$\
> > > where $\max^R_h(U)$  returns the $R$ maximal points (the maximum, the 2nd maximum, \dots, the $R$-th maximum) of $U$ (if $U$ has less than $R$ elements, then the whole $U$ is returned). The key observation is that this algorithm is completely equivalent to a sequential algorithm (where only one point is added at each iteration) that is identical to Algorithm 1 in the paper, except in that $h$ is updated every R iterations:\
> > > &nbsp;&nbsp;&nbsp;&nbsp;&nbsp;&nbsp; 1. Initialize $T =\emptyset$, $h=h_0$, $z= \max_h (D_s)$, $iter=0$\
> > > &nbsp;&nbsp;&nbsp;&nbsp;&nbsp;&nbsp; 2. *while* $z \neq$ Stop, do \
> > > &nbsp;&nbsp;&nbsp;&nbsp;&nbsp;&nbsp; 3. &nbsp;&nbsp;&nbsp;&nbsp;&nbsp;&nbsp; $iter \gets iter+1$\
> > > &nbsp;&nbsp;&nbsp;&nbsp;&nbsp;&nbsp; 4. &nbsp;&nbsp;&nbsp;&nbsp;&nbsp;&nbsp; $T\gets T\cup ${$z$}\
> > > &nbsp;&nbsp;&nbsp;&nbsp;&nbsp;&nbsp; 5. &nbsp;&nbsp;&nbsp;&nbsp;&nbsp;&nbsp; *If* $\mod(iter,R)=0$, *then* $h\gets L([T]_A) $\
> > > &nbsp;&nbsp;&nbsp;&nbsp;&nbsp;&nbsp; 6. &nbsp;&nbsp;&nbsp;&nbsp;&nbsp;&nbsp; $z \gets \max_h (D_s)$\
> > > &nbsp;&nbsp;&nbsp;&nbsp;&nbsp;&nbsp; 7. *end while*\
> > > &nbsp;&nbsp;&nbsp;&nbsp;&nbsp;&nbsp; 8. Return $h,T$\
> > > Illustrating this equivalence is enough to secure the result, because for this sequential algorithm the proof of Theorem 4.2 applies word by word, without any modification.
> > >
> > > * Regarding the gist of the proof of Theorem A.4: To derive the result, in the proof of Theorem A.4 it is provided a characterization of all possible compression schemes in terms of certain probability measures that are needed to evaluate $\mathbb{P}$ {$\underline\varepsilon(\tt{c}(D)|,\delta) \le \phi(D)\le \overline \varepsilon(|\tt{c}(D)|,\delta)$}; then, the maximal value of $\mathbb{P}${$\underline\varepsilon(|\tt{c}(D)|,\delta) \le \phi(D)\le \overline \varepsilon(|\tt{c}(D)|,\delta) $} over the compression schemes characterized as described above is computed, which leads to a non-conservative upper bound  to $\mathbb{P}${$ \underline\varepsilon(|\tt{c}(D)|,\delta) \le \phi(D)\le \overline \varepsilon(|\tt{c}(D)|,\delta)$}. The ensuing maximization problem is infinite dimensional and its solution is obtained by duality. Interestingly, no conservatism is introduced at this stage because strong duality holds. The final expressions for $\underline\varepsilon(|\tt{c}(D)|,\delta)$ and $\overline \varepsilon(|\tt{c}(D)|,\delta)$ given in the statement of Theorem A.4 are obtained by studying the dual problem. See [Campi & Garatti, 2023] for further details.

---

> > > > ### Comment · Reviewer_q61g · 2023-08-19
> > > > **Raising my score to 6**
> > > >
> > > > Thanks for the clarification. Please incorporate all this into the paper. If possible, I would also expand slightly more on the sketch of the proof of Theorem A.4.
> > > >
> > > > I am raising my score to 6.

---

> > > > > ### Author Response · Authors · 2023-08-19
> > > > > **Thanks**
> > > > >
> > > > > Many thanks for your positive opinion and your suggestions, which we believe have helped to improve the manuscript. We will certainly add the material in the final version and expand the paragraph a little more, as you suggested.

---

### Official Review · Reviewer_pkHc · 2023-07-03

**Soundness:** 3 good
**Presentation:** 3 good
**Contribution:** 2 fair
**Rating:** 6
**Confidence:** 3

**Summary:**

This work elaborates on the recent breakthroughs of Campi&Garatti 2023, by exploiting compression theory results to design a novel meta-algorithm, namely the Pick-To-Learn (P2L) algorithm. This algorithm aims to compress the dataset to a smaller, truly impacting one, this notion of impact being defined through a hypothesis dependent order $\leq_h$. Authors provably show high-probability generalisation bounds for P2L and experimentally shows that P2L with gradient descent as subroutine yields better theoretical results and experimental performances than both the test-set approach and PAC-Bayes learning.


In conclusion, I am convinced by the P2L algorithm, and it could be enough for acceptance at this point. However, I remain doubtful about the experimental process, see the Questions part.

**References**
Wu et al. 2022 : Split-kl and PAC-Bayes-split-kl Inequalities for Ternary Random Variables

**Strengths:**

I found the theoretical part of this work utterly interesting as it offers a breath of fresh air in the generalisation field (at least up to my knowledge).

**Weaknesses:**

 I have several concerns about the experimental setup (in particular the comparison with PAC-Bayes theory). See the 'Questions' part.

**Questions:**

- To perform their experiments, authors split MNIST in 60 datasets of size 1000 to enlighten how well works P2L when few data are available, which corresponds to many real-life situations. However, there is other situation where a huge amount of data is available. Thus, I am wondering about the performance of all three methods when trained on all, 60000 data simultaneously. Did the authors try to perform this experiment?
- Appendix B: is the bound 4.a of Clerico et al. truly the tightest? It seems that Wu et al. 2022 improved on this.
- I understand that the current experiments aim to express the interest of the proposed meta algorithm, but I remain doubtful about the comparison with the PAC-Bayes learning objective. Indeed, the comparison with the test method is meaningful as it shows how P2L improves on classical GD. Why not taking the PAC-Bayes minimisation routine as a subroutine of P2L and compare it to PAC-Bayes without P2L (plotted in green in this work)?
- Does it exist similar procedures than P2L in the literature which selects meaningful data?
- What is the time complexity for plotting the bounds and running P2L comparing to other methods?

**Limitations:**

None.

---

> ### Author Rebuttal · Authors · 2023-08-09
>
> First of all, we would like to thank the Reviewer for their valuable time and feedback on the work, which we found useful and to the point.
>
> **Questions**
>
> * “To perform their experiments…”: Considering the small-data regime was motivated both by applications and by the fact that, in these settings holding out a portion of the training set for testing deteriorates the post-training performance.
> At the same time, we completely agree with the Reviewer that there are many learning problems for which large amount of labelled data is available (including MNIST). As suggested, we have performed similar experiments to the case where all three methods are applied the 60000 datapoint simultaneously (employing a pre-training fraction of 0.5), with similar results to those found when using a dataset of size 1000. In particular, the bound / post-training performance achieved are included in the table below.\
> Regarding the bounds: In the present setting, the generalization bound attained by test-set is slightly better compared to that achieved by P2L, and P2L's bound still outperform Pac Bayes’. Regarding the post-training performance: once again P2L and SGD+test-set provide post-training performances that are slightly superior to Pac-Bayes. More precisely, in the present setting P2L slightly outperforms SGD+test-set. These results are due to the fact that, in this regime, leaving out datapoints from the training phase (as done with SGD+test-set and a pre-training fraction of 0.5) has little cost, since the size of the training dataset remains large enough to reach (almost) the optimal model. We aim to clarify these points in final version of the manuscript.
> | Method       | Risk bound | Post-training performance |
> |--------------|------------|---------------------------|
> | P2L          | 2.31%      | 0.99%                     |
> | Pac-Bayes    | 2.54%      | 1.65%                     |
> | SGD+Test-set | 1.65%      | 1.52%                     |
>
> * “Appendix B: is …”: The work of Wu et al. 2020 is certainly interesting, as it provides novel concentration bounds for random variable with ternary values, e.g., with values in {-1,0,1} as opposed to the classical binary setting of {0,1}. At the same time, we note that our work considers an appropriateness criterion that correspond to a binary loss function (either h is appropriate or not). In the binary settings, the same authors observe that their novel bound behaves similarly to existing kl-bounds [Wu et al, lines 5/6 of Introduction, lines 3/5 of Discussion]. Still, we find the work of interest and make sure to mention this point in our final version of the manuscript.
>
> * “I understand that …”: We thank the Reviewer for the insightful comment. We wish to emphasize that the aim of our work is to show that, by using P2L, we can learn a hypothesis that has both post-training performance and risk bounds that are equally good or better compared to state of the art PAC-Bayes or SGD+test-set. In this respect, P2L is not designed with the aim of improving the performance of *any* internal algorithm, but rather that of providing *an* internal algorithm with a generalization bound -- when this is not readily available -- while maintaining a desirable post-training performance. Thus, we believe our comparison to be meaningful and our numerical analysis to support this statement. Certainly, it is interesting to investigate whether P2L can improve the performance of other inner algorithms, in particular P2L+PAC-Bayes, as suggested by the Reviewer. This is a direction we will investigate.
>
> * “Does it exist …”: There certainly is a growing body of literature referred to as “data compression” / “dataset selection” / “coreset selection”, e.g., [A,B,C] that aims at reducing a given dataset’s size while ensuring that the performance of the resulting trained models is comparable. However, these works are different both in their motivation (compressing the dataset is motivated by a computational issue), and in their results. Indeed, none of these works produces a compression scheme in the sense of the classical definition (i.e., such that the model’s output is identical), and certainly they do not enforce the property of *preferent* compression. Both of these properties are crucial to derive a generalization bound such as ours, and we believe our work to be the first -- to the best of our knowledge.
>
>   * [A]: Dataset pruning: reducing training data by examining generalization influence.
>   * [B]: Deep Learning on a Data Diet: Finding Important Examples Early in Training.
>   * [C]: An Empirical Study of Example Forgetting during Deep Neural Network Learning
>
> * “What is the time …”: The following are running times for one instance of MNIST, i.e., a single dataset with 1000 datapoints (pretraining factor =0.5) on the following machine: MBPro 2021, Apple M1 Pro CPU, 32gb ram:
> | Method       | Execution time |
> |--------------|:--------------:|
> | P2L          |     2min 1s    |
> | Pac-Bayes    |     4min 4s    |
> | SGD+Test-set |     0min 5s    |

---

> > ### Comment · Reviewer_pkHc · 2023-08-15
> > **Thank you for your reply**
> >
> > I thank the authors for their careful reply.
> >
> > Most of my concerns are addressed with your rebuttal. I have only a remaining concern about the relevance of comparison in PAC-Bayes. Indeed, I did not understand why you affirm that 'P2L is not designed with the aim of improving the performance of any internal algorithm, but rather that of providing an internal algorithm with a generalization bound'.
> > To me, a meta algorithm is not an internal one. In the context of Algorithm 1, I would call the learning algorithm $L$ an internal procedure. As PAC-Bayes learning aims to provide such a learning algorithm $L$, it seems odd to compare PAC-Bayes to P2L +GD instead of PAC-Bayes with PAC-Bayes + P2L.
> > That being said, I remain an enthusiast about P2L as it is time-efficient and comes with strong theoretical guarantees, but it seems still unfair to me to claim that 'P2L dominates the PAC-Bayes approach' (l.252) as you are comparing a PAC-Bayes learning algorithm with a learning algorithm (GD) enhanced with P2L: P2L and PAC-Bayes are not incompatible, and PAC-Bayes + P2L could solve the fact that in PAC-Bayes, 'the posterior training does not exploit the extra data to improve the model, but rather to certify it.' (l. 260).
> >
> > Did I miss something?
> >
> > PS: Please be sure that I am not necessarily asking for new experiments if those are time-consuming, there are apparent strengths of P2L with respect to the naive PAC-Bayes approach, but to me, it seems strange to compare a learning algorithm with a meta one.

---

> > > ### Author Response · Authors · 2023-08-18
> > >
> > > We would like to thank the Reviewer for the clarification. Our goal was indeed that of comparing P2L+GD against Pac-Bayes and not P2L in itself against Pac-Bayes. Indeed, while P2L is a meta-algorithm, the composition of P2L with GD (P2L+GD) returns a learning algorithm, which then can be compared with other learning algorithms to check what performances P2L+GD offers, both in terms of post-training performance and of enabling generalization bounds. We apologize if this was not conveyed properly.
> > >
> > > Following up on this, our motivation for comparing P2L+GD with PAC-Bayes is that PAC-bayes algorithms have recently pushed the state of the art in terms of being able to learn a good hypothesis while simultaneously certifying it. In this sense, we put P2L+GD at the same level of PAC-Bayes algorithms: both generate a hypothesis and a risk bound, which we are interested in comparing. We also agree with the Reviewer that one could consider running P2L+Pac-Bayes, however we had not considered this as PAC-Bayes algorithms already provide in themselves a risk bound without needing to be used as inner algorithms of P2L.

---

> > > > ### Comment · Reviewer_pkHc · 2023-08-18
> > > >
> > > > Thank you for your reply.
> > > >
> > > > Given the tight generalisation bounds and the fast computational time, I move my score to 6.
> > > > However, I believe that it is unfair to claim that P2L 'dominates PAC-Bayes' as it suggests those two approaches are not compatible. To me, the only way to assess such a compatibility is to plot P2L+PAC-Bayes in the final version and to check whether this leads to improved results (i.e. P2L+PAC-Bayes has the best performances) or not (P2L does not enforce PAC-Bayes and thus P2L is strictly better than PAC-Bayes here).

---

> > > > > ### Author Response · Authors · 2023-08-19
> > > > > **Thanks**
> > > > >
> > > > > Many thanks for your positive judgement and all your advice.
> > > > >
> > > > > Our sentence 'P2L dominates the PAC-Bayes approach' was indeed imprecise, as our intent was to convey that P2L+GD (and not P2L in itself) outdoes the PAC-Bayes approach in the considered experiment. We apologize for not conveying this properly and will modify the sentence. We also agree with the Reviewer that the comparison in the suggested form is worth trying. Before the final version, we shall run P2L + PAC-Bayes and provide the corresponding plot in the supplementary material. This additional simulation will certainly add value to our paper and better position it within the existing methods.

---

### Official Review · Reviewer_3uwv · 2023-07-06

**Soundness:** 3 good
**Presentation:** 3 good
**Contribution:** 3 good
**Rating:** 7
**Confidence:** 3

**Summary:**

In this paper, the authors present a novel framework called P2L, which aims to derive generalization guarantees for black-box supervised learning algorithms. P2L operates as a meta-algorithm that utilizes a learning algorithm to induce a compression scheme. The algorithm relies on two main components: a *criterion of appropriateness* and an *appropriateness threshold*. The criterion of appropriateness serves as a generalization of the loss function, measuring how well a hypothesis describes a given data point. The appropriateness threshold determines when the meta-algorithm terminates by ensuring that the hypothesis is appropriate for all unselected data points. At each step, the algorithm selects the example with the highest loss, according to the criterion of appropriateness, from the set of selected training data ($T$). This example, denoted as $\bar{z}$, is added to $T$, and the learning algorithm produces a hypothesis ($h$) based on $T$. Using this new hypothesis, a new $\bar{z}$ is selected, and the process iterates until h becomes appropriate for all examples in the original dataset ($D_S$). An important property of P2L is that running the algorithm on $T$ yields the same hypothesis as running it on the full dataset $D_S$. Consequently, $T$ effectively compresses $D_S$, similar in spirit to dataset distillation or identifying core examples within a dataset. To provide generalization guarantees, a theorem demonstrates that the cardinality of $T$ can be utilized to derive tight upper bounds on a suitable measure of statistical risk. The effectiveness of P2L is evaluated through experiments on binary MNIST and synthetic regression datasets. The results indicate that P2L outperforms the PAC-Bayes bound and performs competitively with using a hold-out set, all without requiring additional data.

**Strengths:**

- To the best of my knowledge, the proposed algorithm is novel and quite elegant (computational complexity notwithstanding). It is another realization of the principle “compression is intelligence”.
- I find the technical tools used interesting and believe that this framework can potentially lead to many future works.
- The resulting bounds are tighter than PAC-Bayes bounds.
- The presentation of the algorithm is very clean and easy to follow. Most technical concepts are explained carefully and illustrated with examples. This is quite rare for theory papers.
- The algorithm exhibits some very interesting properties that are potentially desirable for learning algorithms. (*"the misclassification on the test-set for P2L is constant across all prior/train portions"*)


**Weaknesses:**

- In contrast to many existing generalization bounds, P2L can only be used for risk certification, that is, it does not offer an immediate way to make the bound tighter (which may improve the performance of the model) nor does it provide an understanding of why particular algorithms or architecture work well. Of course, this is partially due to the fact that P2L is designed to be as general as possible (i.e., black-box) but depending on the goal, the latter is sometimes more important than the risk certification itself.
- The algorithm seems not particularly efficient which limits its practicality, especially for deep learning with very large datasets. If the model is trained from scratch at every iteration, the complexity could be quadratic in the number of data points (i.e., $N^2$). On the other hand, it seems more suited for applications where the dataset is small. Perhaps it's better to phrase the paper in that direction. If this is not the case, please correct me. The authors do already address this point in the conclusion but note that PAC-Bayes generalization bounds for deep learning often use Gaussian posterior which is easy to sample from and empirically the estimate concentrates quite fast. Furthermore, these bounds can be made deterministic (Nagarajan et al., 2018).
- Related to the previous point, the paper would benefit from more empirical evaluation such as those in (Lotfi et al., 2022) who provided the SOTA PAC-Bayes bounds for several benchmarks. The field of PAC-Bayes bounds for deep learning has made significant progress in the past couple of years and only having binary MNIST results makes it hard to judge the empirical value of the proposed algorithm.
- The paper is generally well-written and clear but there are several important clarifications needed (see questions).

**Reference**

- PAC-Bayes Compression Bounds So Tight That They Can Explain Generalization. Lotfi et al., 2022
- Deterministic PAC-Bayesian generalization bounds for deep networks via generalizing noise-resilience. Nagarajan et al., 2018

**Questions:**

- The role of $h_0$ is somewhat unclear to me from the text. What does "*pre-training $h_0$ on different portions of the dataset, again through GD*" mean (line 235)? Why is the model not initialized from scratch? In algorithm 1, where is $h_0$ used? Is it only used for selecting the initial $\bar{z}$?
- Why did you use GD instead of SGD? Wouldn’t this result in worse models? How do the bounds compare if you use SGD?
- In the inner loop, does the training have to be done from scratch or can it rely on the hypothesis from the previous iteration? If it is done from scratch, it feels like the algorithm would be extremely impractical since every iteration has to train a new neural network. If it is done starting from the weights from the previous iteration, wouldn’t the weights be prone to overfitting, especially if the problem is non-convex?
- Is post-training performance just the test performance? If so, could you elaborate on the following statement “*utilizes all data to jointly learn a good model and provide a risk bound*"? Do you have a theoretical justification for this statement? It feels like this could have some relationship to curriculum learning or boosting.
- Is there a relationship between $\gamma$ and the notion of margin? Why does $\gamma$ not show up in the bound? Is the dependency implicit in $T$?
- Is the total ordering determined by the most recent hypothesis $h$ or the meta-algorithm? There seem to be some conflicting statements in the paper. “$[T]_A$ *is a list that contains the elements in* $T$ *ordered according to the order in which they are selected by P2L*” (line 150) but in the conclusion *“a hypothesis-dependent total order used to select which data points are fed to the learning algorithm L.”* (line 346). In a similar vein, how is the ordering used in the experiment since GD does not care about the order of the data at all?
- Does this framework have a relationship to the algorithmic stability framework?

**Limitations:**

See the weakness section.

---

> ### Author Rebuttal · Authors · 2023-08-09
>
> We would like to thank the Reviewer for their positive review and constructive comments. We include our responses below, which we hope can help better understand and position our paper.
>
> **Weaknesses**
>
> * “In contrast…”: while it is true that our bound cannot be directly optimized, the size of the compression set T (and the corresponding bound on the risk) are informative indicators of whether the chosen architecture had worked well. When this is not the case, the flexibility of P2L allows to accommodate and test a different inner algorithm with the goal of improving the result.
>
> * “The algorithm seems…”: we agree with the Reviewer that P2L is, in its current form, more suited to offline problems with dataset of relatively small size – a point that we will emphasize more. However, please also note that, in situations where P2L works well (i.e. a small size of the compression is achieved), P2L halts at early stages, during which the inner algorithm can be executed efficiently since it is run over a smaller portion of the dataset.
>
> **Questions**
>
> * “The role of $h_0$…”: As you pointed out, $h_0$ is used to select the first $\bar{z}$ to be put in T. From that step on, $h$ is computed via the inner learning algorithm fed with T. As natural, note that $h_0$ has an impact on the algorithm: if $h_0$ is already good, then the algorithm just needs to refine it and will likely terminate at early stages with a small size of the compression set; if $h_0$ is poor, P2L may need some additional iterations to construct a set T from which to obtain a sensible hypothesis. In this respect, pre-training is introduced solely to guide the selection of $h_0$. In particular, a portion of the training dataset is used to generate a sensible $h_0$ to start from, and then P2L is used on the remaining data. We apologize if this was not clear and will work to better explain this point in the final version of the paper.
>
> * “Why did you use GD…”: We tried both GD and SGD and they gave similar results. We will mention this in the paper.
>
> * “In the inner…”: the generality of P2L allows one to choose either of the following approaches: one can either re-train a hypothesis from scratch at every iteration or keep the previously obtained hypothesis as initialization (this was our choice in the simulations). The two implementations can be seen as slightly different inner learning algorithms for P2L. Interestingly, the certification on the risk that we provide allows one to understand whether overfitting has arisen (in which case the risk certification will turn out to be close to 1) and therefore to opt for a different approach, or inner algorithm altogether.
>
> * “Is post-training…”: yes, by post-training performance we mean the performance on a test dataset. To be precise, we use a test dataset to assess the actual performance of the three methods (e.g., 10000 examples for MNIST), in addition to the dataset D employed by each of the methods. At the same time, we note that, while P2L and PAC-Bayes are allowed to utilize the whole dataset D to optimize the hypothesis, SGD+test-set is forced to further divide D in D1 and D2 where D1 is used for training and D2 to return a certification of the risk. The sentence you mentioned is motivated by the experimentally verified fact (see Figure 2) that P2L always obtains an actual performance (as measured on the hold-out dataset) which coincides with that obtained by SGD+test-set when one takes D1 = D (the “good model” in the sentence). For this model, SGD+test-set cannot provide any risk certification since D2 is empty, while P2L instead is able to provide meaningful risk bounds. We apologize if this was not clear and aim to further discuss this point in the final version of the paper.
>
> * “Is there a relationship...”: within our setting, the “margin” is the difference between gamma and the distance of the worst example from the chosen model. As correctly pointed out by the Reviewer, the risk depends on gamma, but gamma does not appear in the bound because the dependence is implicit in T. Indeed, T and the compression size depend on the chosen value of gamma: for example, if gamma is selected to be very large, the corresponding compression T will likely be small because a coarse model will be sufficient to accommodate the points within a (large) threshold of gamma.
>
> * “Is the total ordering…”: there are two different ordering coming into play here, and we apologize if this was not clearly conveyed. On the one hand, we have an ordering relation over the datapoints in $D_s\setminus T$ that depends on $h$, and which is used to select the worst example for the present $h$ to augment T. On the other hand, the order of the elements in the list $[T]_A$ simply refers to the position of the elements in the list and is solely used to enable inner learning algorithms that depend on the order with which they are fed with data, e.g., the output of SGD depends on the order with which the points in the dataset T appear. In case of order-of-feeding independent algorithms, like e.g. GD, the order of the elements in $[T]_A$  is simply ignored by the algorithm and has no effect on the final result.
>
> * “Does this framework…”: While the framework is not related to the algorithmic stability framework of [A], the notion of stable compression schemes (introduced by [B] and recently reconsidered in [C]), which is equivalent to the notion of preferent compression scheme, is instead central. Many thanks for the question, we will mention this equivalence in the paper.
>
> **References**
> * [A] Bousquet & Elisseff. Stability and Generalization, JMLR, 2022
> * [B] V. Vapnik and A. Chervonenkis. Theory of Pattern Recognition, 1974
> * [C] Hanneke & Kontorovich, Stable sample compression schemes: new applications and an optimal SVM margin bound. PMLR, 2021

---

> > ### Comment · Reviewer_3uwv · 2023-08-15
> >
> > Thank you for the detailed response. Most of my concerns have been addressed. I intend to keep my current rating. I have one more question:
> >
> > > the certification on the risk that we provide allows one to understand whether overfitting has arisen
> >
> > Can you elaborate on this point? And also can you discuss the trade-off of retraining from scratch vs not in the inner loop?

---

> > > ### Author Response · Authors · 2023-08-18
> > >
> > > * Regarding diagnosing overfitting: The point we wished to convey here — perhaps too concisely — is that, by providing the final hypothesis with a generalization guarantee, our risk bound also indirectly informs on how effective the selection of model and the training procedure have been. This, in turn, can help diagnose instances in which overfitting has arisen. For example, instances in which the loss on the training dataset is small, but the generalization risk provided by our bound is high are immediate candidates for overfitting. We thank the Reviewer and will mention this point in the final version of the manuscript.
> > >
> > > * Regarding retraining from scratch vs not: This is another interesting point, and indeed there is a potential tradeoff to be uncovered. On one hand, the rational for not retraining from scratch at every iteration is twofold. First, there is a potential computational advantage as fewer gradients steps are often needed to update the hypothesis since we start from a reasonably good one already. Second, such an approach often allows for the hypothesis $h_k$ to be less sensitive to the addition of one datapoint. This is typically helpful in the last stages of the algorithms where there isn’t much to be learned and the main goal is that of terminating quickly, a task which would be more difficult if the hypothesis was to change significantly. On the other hand, re-training from scratch also has advantages, mostly in that the algorithm is given full flexibility in moving towards a better hypothesis, an aspect that is often convenient in the first phases of P2L. Overall, we have tested both solutions and found that, for the problems considered, the first approach has an advantage over the second.

---

### Author Rebuttal · Authors · 2023-08-09

We would like to take this opportunity to thank all the Reviewers and PCs for their valuable time and feedback. Below, we address each of the reviews individually, while we attach here a pdf containing additional figures used in the responses.

---

### Decision · Program_Chairs · 2023-09-21

**Decision:**

Accept (spotlight)

**Comment:**

A very well-written paper that proposes a novel and elegant method for obtaining generalizing guarantees for black box models. The main idea is to propose a modification to the traditional learning algorithm in a way that leads to data compression. Then some recent work on data compression is exploited to obtain guarantees. This does not require any prior knowledge of the learning algorithms. This paper was well received. The results are correct and clearly stated. Reviewer q61g highlighted that the proofs are not very complex and rely on some previous work. However this is not a very big criticism as the main contribution is not theoretical but rather methodological and conceptual. I  recommend this paper to be accepted as a spotlight.